# Towards A Unified View of Sparse Feed-Forward Network in Pretraining Large Language Model

**Leo Z. Liu**♠♡ *    **Tim Dettmers**♠    **Xi Victoria Lin**♡    **Veselin Stoyanov**♡    **Xian Li**♡

♠University of Washington, Seattle ♡Meta AI.

## Abstract

Large and sparse feed-forward layers (S-FFN) such as Mixture-of-Experts (MoE) have proven effective in scaling up Transformers model size for *pretraining* large language models. By only activating part of the FFN parameters conditioning on input, S-FFN improves generalization performance while keeping training and inference costs (in FLOPs) fixed. In this work, we analyzed two major design choices of S-FFN: the memory block (a.k.a. expert) size and the memory block selection method under a general conceptual framework of sparse neural memory. Using this unified framework, we compare several S-FFN architectures for language modeling and provide insights into their relative efficacy and efficiency. We found a simpler selection method — `Avg-K` that selects blocks through their mean aggregated hidden states, achieving lower perplexity in language model pretraining compared to existing MoE architectures including Switch Transformer (Fedus et al., 2021) and HashLayer (Roller et al., 2021).

## 1 Introduction

Large-scale pretrained language models (LLMs) achieve remarkable performance and generalization ability for NLP tasks (Radford and Narasimhan, 2018; Devlin et al., 2019; Liu et al., 2019; Radford et al., 2019; Brown et al., 2020; Raffel et al., 2020; Chowdhery et al., 2022). Scaling up the model size (the number of parameters) has been shown as a reliable recipe for better generalization, unlocking new capabilities, while the performance has not shown signs of plateauing (Kaplan et al., 2020; Zhang et al., 2022a; Chowdhery et al., 2022; Hoffmann et al., 2022; Wei et al., 2022). However, the computational resources required to train larger language models are formidable, calling for more efficient training and inference solutions of LLMs (Borgeaud et al., 2022; Schwartz et al., 2020; Tay et al., 2020).

One promising direction is sparse scaling which increases the number of parameters while keeping the training and inference cost (in FLOPs) fixed. Recent work focuses on scaling up a transformer's feed-forward network (FFN) with sparsely activated parameters, resulting in a scaled and sparse FFN (S-FFN). There have been two major approaches to achieve S-FFN. One treats S-FFN as a neural memory (Sukhbaatar et al., 2015a) where a sparse memory retrieves and activates only parts of the memory cells (Lample et al., 2019). The other adopts Mixture-of-Expert Network (MoE) (Lepikhin et al., 2021; Fedus et al., 2021; Du et al., 2021; Roller et al., 2021; Lewis et al., 2021; Chi et al., 2022) that replaces a single FFN module with multiple equal-sized ones (called "experts") and only activates a few among many experts for a particular input.

While both memory and MoE models achieve S-FFN, they have been considered two distinct approaches. We aim to draw the connections between these two classes of S-FFN: What critical design choices do they have in common? Which design choices are essential for their modeling capability and computation efficiency? Can the effective ingredients of each method be transferred and combined to improve performance further?

In order to answer these questions, we start from the neural memory view of FFN (Sukhbaatar et al., 2015a) (§2.1) and reduce all S-FFN's to the same mathematical form (§3.1). Then, we characterize these methods along two dimensions — memory block size (e.g. expert size) and memory block selection method (e.g. gating) (§3.2). Using this framework, we made the following contributions:

- We study a wider range of memory block sizes besides the commonly used block size in MoE architectures (Fedus et al., 2022) and show that reducing the block size keeps improving

---

*Work done as a Meta AI Resident. Correspondence to: Leo Zeyu Liu <zeyuliu2@cs.washington.edu>

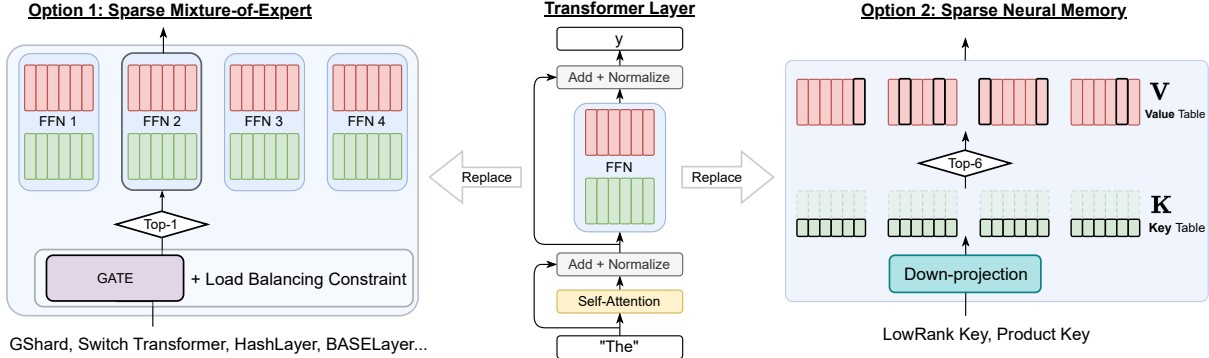

Figure 1: Sparse Mixture-of-Expert and Sparse Neural Memory as two different methods.

the perplexity with little incurred extra FLOPs (§5.1 §5.2), leading to better perplexity/computation trade-offs.

- We conduct a systematic exploration of block selection methods to quantify their relative efficacy and efficiency (§5.2). Specifically, we find that the selection method through a gating function, in general, improves the FLOPs-Perplexity trade-off. However, the parameterization of the current MoE's (Fedus et al., 2021; Lewis et al., 2021) gating function — multiplying token representation with a separately learned matrix — has worse perplexity than using the FFN hidden states — multiplying token representation with FFN's matrix (§2.1).

- Drawing on the insights above, we propose a simple gate for S-FFN— **Avg-K** (§3.3) as a hybrid design choice between sparse neural memory and a mixture of experts. It efficiently selects memory blocks based on the mean aggregated hidden states of each block. With 1% additional FLOPs, **Avg-K** achieves lower perplexity (14.80) on the validation set of The Pile (Gao et al., 2020) than both Switch Transformer (16.45) (Fedus et al., 2021) and HashLayer (15.75) (Roller et al., 2021). Moreover, **Avg-K** is the first MoE model that performs well without load balancing constraint, while conventional MoE transformers like Switch Transformer will degenerate (Fedus et al., 2021; Shazeer et al., 2017; Eigen et al., 2014).

## 2 Background

### 2.1 Feed-Forward Network

A transformer layer (Vaswani et al., 2017) consists of a self-attention block and a Feed-Forward Network block (FFN). FFN receives an input vector $\mathbf{x} \in \mathbb{R}^d$ from the self-attention block, multiplies it with parameter matrix $\mathbf{K} \in \mathbb{R}^{d_m \times d}$, applies a non-linear function $f$ to obtain the hidden states $\mathbf{m} \in \mathbb{R}^{d_m}$ and applies another affine transformation $\mathbf{V} \in \mathbb{R}^{d_m \times d}$ to produce a $d$-dimensional output $\mathbf{y}$. This multi-layer perceptron architecture can be expressed as:

$$\mathbf{y} = \text{FFN}(\mathbf{x}) = f(\mathbf{x} \cdot \mathbf{K}^\top) \cdot \mathbf{V} = \mathbf{m} \cdot \mathbf{V}. \quad (1)$$

Additionally, we could also view it as a neural memory (Sukhbaatar et al., 2015b, 2019; Geva et al., 2021) (Eq. 2).

$$\mathbf{y} = \sum_{i=0}^{d_m-1} f(\mathbf{x} \cdot \mathbf{k}_i) \cdot \mathbf{v}_i = \sum_{i=0}^{d_m-1} m_i \cdot \mathbf{v}_i. \quad (2)$$

In this view, FFN consists of $d_m$ key-value pairs, known as *memory cells*. Each key is represented by a $d$-dimensional $\mathbf{k}_i \in \mathbb{R}^d$, and together form the *key table*; likewise, value vectors $\mathbf{v}_i \in \mathbb{R}^d$ constitutes a *value table* $\mathbf{V} \in \mathbb{R}^{d_m \times d}$. The memory multiplies the query input $\mathbf{x} \in \mathbb{R}^d$ with every $\mathbf{k}_i$; followed by the non-linear function, it produces *memory coefficient* $m_i = f(\mathbf{x} \cdot \mathbf{k}_i)$ for the $i$-th memory cell. Finally, the output of FFN is the sum of its values $\mathbf{v}_i$ weighted by their corresponding memory coefficient $m_i$. Conventionally, the *size of FFN* — $d_m$ — is set to be $4 \cdot d$.

### 2.2 Scaling up FFN

As discussed in §1, scaling up the number of parameters in FFN serves as a lever to improve trans-

former performance. Since a standard FFN accounts for about two-thirds of a transformer layer's parameters (Geva et al., 2021), scaling up FFN will greatly affect the parameter size of a transformer model. However, one could sparsely activate the parameter to control the required compute. In this section, we and discuss two approaches to achieve a *scaled* and *sparse* FFN (S-FFN). One has a mixture-of-expert model activate a few experts (§2.2.1), and the other specifies a memory model to sparsify (§2.2.2).

### 2.2.1 Mixture of Experts (MoE)

Mixture of experts (MoE; Jacobs et al. (1991)) consists of a set of expert models $\{\mathbf{f}_i(\mathbf{x})\}_{i=0}^{B-1}$ and a gating function $\mathbf{g} : \mathbb{R}^d \to \mathbb{R}^B$ to estimates the relevance of each expert. Finally, the output is the sum of experts' output weighted by the gate's weight estimation for that particular expert.

$$\text{MoE}(\mathbf{x}) = \sum_{i \in \mathcal{E} = \{0,1,\cdots,B-1\}} \mathbf{g}_i(\mathbf{x}) \cdot \mathbf{f}_i(\mathbf{x}) \quad (3)$$

Recent work (Du et al., 2021; Lepikhin et al., 2021; Roller et al., 2021; Lewis et al., 2021; Zhou et al., 2022) have applied this approach to transformer by dissecting the FFN into multiple expert blocks and sparse activation of the **MoE** (**SMoE**). In SMoE, the gating function (or "router") routes an input token $\mathbf{x}$ to a subset[1] (e.g. 1 or 2) of experts, $\mathcal{E} = \text{subset}(\mathbf{g}(\mathbf{x}))$. Previous work mainly adopts two types of gates.

**Learned gate** is parameterized by a set of learnable expert embeddings $\boldsymbol{\theta} = [\mathbf{e}_0; \cdots ; \mathbf{e}_{B-1}] \in \mathbb{R}^{B \times d}$, where each embedding corresponds to one expert. The relevance of the $i$-th expert is obtained by

$$\mathbf{g}_i(\mathbf{x}) = \frac{\exp\left(\mathbf{e}_i \cdot \mathbf{x}\right)}{\sum_j \exp\left(\mathbf{e}_j \cdot \mathbf{x}\right)}$$

To enforce load balancing when routing, previous work have employed an additional auxiliary loss (Lepikhin et al., 2021; Fedus et al., 2021; Artetxe et al., 2021; Du et al., 2021; Chi et al., 2022) or framed expert utilization as a constrained optimization problem (Lewis et al., 2021; Zhou et al., 2022).

---

[1]Conventionally, SMoE implements **load balancing** constraints to prevent the overuse of certain experts and the underutilization of others, and avoid convergence to local optima (Shazeer et al., 2017; Eigen et al., 2014).

**Static gate**, in contrast to a learnable gate, does not have any differentiable parameters. Instead, it uses a static mapping that encodes load-balancing constraints to route input (Roller et al., 2021; Gururangan et al., 2021). For example, RandHash from HashLayer (Roller et al., 2021) uses a hash table that maps from token type to *randomly* selected expert(s). DEMix (Gururangan et al., 2021) ensures each expert only sees data from a pre-defined domain.

### 2.2.2 Sparse Neural Memory

The other line of work follows the memory view of FFN (Eq. 2). It is straightforward to increase the memory size $d_m$ to a much larger value $d_m \gg 4 \cdot d$. By only using the top-$k$ entries in the memory coefficient $\mathbf{m} = \mathbf{x} \cdot \mathbf{K}^\top$ (Eq. 2), one could sparsely activate the value table, resulting in a vanilla sparse memory (**VanillaM**). However, the naive implementation of this approach results in computation cost proportional *linearly* to the memory size. Lample et al. (2019) explored the following two techniques in this direction to scale computation sublinearly.

**Low-Rank Key Memory (LoRKM)** A straightforward technique is to assume that the full key table $\mathbf{K}^\top \in \mathbb{R}^{d \times d_m}$ is composed of and approximated by a downward projection $\mathbf{D} \in \mathbb{R}^{d \times d_\ell}$ and a low-rank key table $\tilde{\mathbf{K}} \in \mathbb{R}^{d_m \times d_\ell}$,

$$\mathbf{K}^\top = \mathbf{D} \cdot \tilde{\mathbf{K}}^\top$$

where $d_\ell \ll d$. LoRKM produces memory coefficients by $m_i = f((\mathbf{x} \cdot \mathbf{D}) \cdot \tilde{\mathbf{k}}_i) = f(\mathbf{t} \cdot \tilde{\mathbf{k}}_i)$.

**Product Key Memory (PKM)** Building upon LoRKM, PKM further decomposes the low-rank key table by assuming different low-rank keys have structured sharing with each other. See Appendix B for more technical details. Due to such factorization, PKM has a negligible key table $\mathbf{K}^\top = \mathbf{D} \cdot \tilde{\mathbf{K}}^\top$ (e.g., $< 0.3\%$) relative to the parameters in the value table.

## 3 A Unified View of Sparse FFNs

We show the connections between MoE and neural memory despite their different formulations on the surface. We first derive a variant form of MoE to establish its connection with sparse memory (§3.1). Then, we propose a unified framework for S-FFN (§3.2).

### 3.1 A Neural Memory View of MoE

MoEs use a gating function to estimate the importance of all experts and combine each expert's output through linear combination. Here, inspired by the memory view on FFNs (§2), we could view MoE as a wide neural memory chunked into $B$ FFNs:

$$
\begin{aligned}
\text{MoE}(\mathbf{x}) &= \sum_{i=0}^{B-1} \mathbf{g}_i(\mathbf{x}) \cdot \text{FFN}^{(i)}(\mathbf{x}) \\
&= \sum_{i=0}^{B-1} \mathbf{g}_i(\mathbf{x}) \cdot \left( \sum_{j=0}^{d_m-1} m_j^{(i)} \cdot \mathbf{v}_j^{(i)} \right) \\
&= \sum_{i=0}^{B-1} \sum_{j=0}^{d_m-1} \left( \mathbf{g}_i(\mathbf{x}) \cdot m_j^{(i)} \right) \cdot \mathbf{v}_j^{(i)} \\
&= \sum_{l=0 \text{ s.t. } l=i\cdot d_m+j}^{B\cdot d_m-1} \left( \mathbf{g}_i(\mathbf{x}) \cdot m_j^{(i)} \right) \cdot \mathbf{v}_j^{(i)} \\
&= \sum_{l=0}^{B\cdot d_m-1} m_l \cdot \mathbf{v}_l, \qquad\qquad (4)
\end{aligned}
$$

where $\text{FFN}^{(i)}$ denotes the $i$-th FFN.

Based on linear algebra, we are able to write the standard MoE formulation (§2.2.1) in a similar summation form as that of neural memory (Eq 4; 2.1). MoE in its neural memory form has a value table $\mathbf{V} = \left[ \mathbf{V}^{(0)}; \cdots ; \mathbf{V}^{(B-1)} \right] \in \mathbb{R}^{B\cdot d_m \times d}$ — the concatenation of $B$ value tables $\mathbf{V}^{(i)} \in \mathbb{R}^{d_m \times d}$ from all $\text{FFN}^{(i)}$. In this view, $\mathbf{v}_l$ with $l = i \cdot d_m + j$ corresponds to the $j$-th value vector in the $i$-th chunk of value table $\mathbf{V}^{(i)}$. Thus, its corresponding memory coefficient $m_l = \mathbf{g}_i(\mathbf{x}) \cdot m_j^{(i)}$ is produced by weighting the $j$-th memory coefficient of $\text{FFN}^{(i)}$, $m_j^{(i)} = \mathbf{x} \cdot \mathbf{k}_j^{(i)}$, by the relevance score of $\text{FFN}^{(i)}$, $\mathbf{g}_i(\mathbf{x})$. Through this memory view, one could see that a *sparse* $\text{MoE}(\mathbf{x})$ is a *sparse* memory operating in terms of $B$ memory blocks; it uses the gate $\mathbf{g}(\mathbf{x})$ to narrow down the calculation over the stacked value tables $\mathbf{V}$ to the value tables from $\text{FFN}^{(i)}$, for $i \in \text{subset}(\mathbf{g}(\mathbf{x}))$ (i.e. sparsify).

**Comparison with Sparse Memory** Both SMoE and sparse neural memory are neural memory, but there are several differences: **1)** whether memory cells share the same relevance weight: in sparse neural memory, each memory cell receives an individual weight $m_i$. In contrast, in SMoE, each group of $4 \cdot d$ memory cells shares the same relevance weight $\mathbf{g}_i(\mathbf{x})$. **2)** memory selection criterion: if we center the key-value computation (Eq. 2) at the core of S-FFN computation, the sparse memory directly uses the memory parameters for selection — the dot product between input token $\mathbf{x}$ and key vectors $\mathbf{k}_i$, whereas SMoE depends on a separately parameterized gate $\mathbf{g}$.

### 3.2 The Unified Framework

We propose a general framework that unifies the two different approaches to achieve S-FFN. We view both as instances of a memory with large key and value table — $\mathbf{K} \in \mathbb{R}^{d_m \times d_k}, \mathbf{V} \in \mathbb{R}^{d_m \times d_v}$, where $d_m \gg 4 \cdot d$. We distinguish the different methods along two dimensions illustrated below and summarized in Table 1:

**Memory block size** specifies how many memory cells share the same relevance weight at selection time, and thus together treated as a memory *block*. We use $g$ to denote the size of one block. In other words, we split the $\mathbf{K}, \mathbf{V}$ along the $d_m$-dimension into $g$-size blocks. Therefore, a memory consists of $B = d_m/g$ blocks in total. Formally, we write

$$
\begin{aligned}
\mathbf{K}^g &= \left[ \mathbf{K}^{(0)}; \mathbf{K}^{(1)}; \cdots ; \mathbf{K}^{(B-1)} \right] \in \mathbb{R}^{d_m \times d_k} \\
\mathbf{V}^g &= \left[ \mathbf{V}^{(0)}; \mathbf{V}^{(1)}; \cdots ; \mathbf{V}^{(B-1)} \right] \in \mathbb{R}^{d_m \times d_v}
\end{aligned}
$$

For example, sparse memory has block size $g = 1$ — trivially treating 1 memory cell as a "block"; and SMoE has the block size $g = 4 \cdot d$ (§3.1). Current approaches generally use fixed block sizes, but this is mostly an artifact of how the methods were derived rather than a mathematical constraint. For example, we can design SMoE versions instead of 1 expert of size $4 \cdot d$, or uses 2 experts of size $2 \cdot d$. We can similarly chunk memory coefficients $\mathbf{m}$ into blocks of size $g$ in sparse memories.

**Memory block selection method** is the specific function that compute the relevance of each memory blocks for selection. Since SMoE is also a type of sparse memory, we distinguish the selection method by a new criterion — *whether one allows input $\mathbf{x}$ to directly interact with the key table $\mathbf{K}^g$*. As discussed in §3.1, SMoE uses the estimation from an individually parameterized gate to select, while sparse memory solely and directly uses a key table. Thus, current SMoE is a type of *indirect* selection method, and sparse memory a *direct* one. Various SMoEs are further characterized by whether their gating function has learned parameters or consists of a static mapping (§2.2.1).

| Memory block size ($g$) | Memory block selection method | | Model Name |
|---|---|---|---|
| 1 | Direct | Full-parameter Key | VanillaM |
| | | Low-rank Key | LoRKM, PKM (Lample et al., 2019) |
| $4 \cdot d$ | Indirect | Learned gate | Switch Transformer(Fedus et al., 2021), GShard (Lepikhin et al., 2021), GLaM (Du et al., 2021), BASELayer (Lewis et al., 2021), X-MoE (Chi et al., 2022) |
| | | Static gate | HashLayer(Roller et al., 2021), DEMix (Gururangan et al., 2021) |

Table 1: S-FFN methods decomposed along the defined design dimensions.

Meanwhile, sparse memory is characterized by how much factorization the key table uses (§2.2.2).

### 3.3 A New Selection Method — `Avg-K`

The gate design in the Mixture-of-Expert methods ensures that not all experts are activated for routing tokens. Without the gate design, conventional sparse neural memory methods (§2.2.2) uses the full key table before sparsifying the computation in the value tables, which explains the increasing computation cost for scaling up sparse neural memory. As later shown in Fig. 3 (§5.1), this additional computation doesn't bring proportional improvement to its performance, compared with MoE.

However, there is still merit in sparse neural memory that MoE could acquire. Based on the contrastive analysis from §5.2, we found that when the model *makes more use of each expert's key table* for routing tokens, the more its performance will increase. This is precisely the deficiency in the conventional Mixture of Experts (MoE) model, as it relies on a separately learned parameter or a static gating mechanism (§2.2.1).

To this end, we propose a new routing method — `Avg-K`— as a hybrid design choice between sparse neural memory and MoE methods. `Avg-K` represents each block with the average of its key table $\mathbf{K}^{(i)} \in \mathbb{R}^{g \times d}$ along $g$-dimension:

$$\mathbf{e}_i = \frac{1}{g} \cdot \sum_{j=0}^{g-1} \mathbf{k}_j^{(i)} = \text{Avg}(\mathbf{K}^{(i)}, \text{ dim=0})$$

Then, we use the dot product between $\mathbf{x}$ and the averages to select the top-$b$ selected block and route the token there for memory calculation (Eq. 2):

$$\mathbf{g}_i(\mathbf{x}) = \begin{cases} 1 & i \in \{\text{top-}b \text{ of } [\mathbf{e}_0, \cdots, \mathbf{e}_{B-1}] \cdot \mathbf{x}\} \\ 0 & \text{otherwise} \end{cases}$$

Due to the linearity of averaging, the operation $\mathbf{e}_i \cdot \mathbf{x}$ is equivalent to calculating the average of dot products within a block without GeLU. Since all tokens share the set of averaged key vectors, our method is efficient. In summary, `Avg-K` migrates MoE's advantages: a gate design, and a full-parameterized key table in each expert. As an advantage of sparse memory, it uses the average key vectors of each expert to route tokens and thus increases the memory selection method's dependency on the key table. We provide more rationale for our choice of average function in Appendix D.1.

## 4 Experiment Setup

### 4.1 Models

We choose **Dense Baseline** using transformer architectures used in GPT-3 models (Brown et al., 2020), which has 24 transformer layers, with $d = 1024$, $f = \text{GeLU}$ as activation function, and with a memory size (or FFN hidden size) to be $4 \cdot d$. This model is also referred to as the "base model" because S-FFN's size is chosen based on the configuration of this model. Our choice of architecture size leads to a base model with 355M parameters (See Appendix A). Our reason for the chosen model size is two-fold: 1) this size is similar to the community-estimated size of OpenAI `text-ada-001`; 2) as indicated in Clark et al. (2022), 355M is the smallest size that separates the performances of different architecture designs.

**S-FFN** Given a model above, we replace some of its FFNs with an S-FFN. Similar to (Lepikhin et al., 2021), we replace the FFN at every 6 layers (layer 5, 11, 17, 23, indexed from 0), leading to 4 S-FFNs in total across 24 layers. We use $k$ to denote the number of memory blocks used and control how

activated the S-FFN is. We use the formulation of $d_m = E \cdot (4 \cdot d)$ to control the size of S-FFN, so the S-FFN will activate $b = \frac{k}{g}$ out of $B = \frac{d_m}{g}$ memory blocks. In Table 2, we list all S-FFN models used for analysis in §5. We count FLOPs analytically following Narayanan et al. (2021) and do not account if a worker finishes computation before another (when using model parallelism). We use *the number of learnable parameters* to consider whether two models are equally expressive. In Table 2, we list all S-FFN models used for analysis in §5. For experimentation of our `Avg-K`, we don't enforce load balancing for simplicity.

**PKM-FFN** Since the factorized key table in PKM has little ($< 0.3\%$) learnable parameter relative to the value table, we propose an indirect variant called PKM-FFN to match the number of parameters of other models like RandHash. This variant has memory block size $g = 1$ and the same key-value table as RandHash. PKM-FFN has a gate whose $\mathbf{g}(\mathbf{x})$ is the same as the $\mathbf{m}$ from a PKM and $\mathbf{g}_i = m_i$; and no load-balancing is enforced.

## 4.2 Language Modeling

**Pretraining Data** We pretrain all S-FFN models on a total of 453GB text with 112B tokens from a union of six English-only datasets, including English subset of CC100 and the five datasets used to pretrain RoBERTa (Liu et al., 2019) — specifically BookCorpus, English Wikipedia, CC-News, OpenWebText, CC-Stories (details in Appendix A.3). We adopt the same Byte-Pair Encoding as GPT-2 (Radford et al., 2019) and RoBERTa (Liu et al., 2019) with a vocabulary of 50K subword units. All models are trained for 60B tokens.

**Evaluation settings** We evaluate our models' ability to predict the next token in a sequence as measured by perplexity. We report both **in-domain** and **out-of-domain** perplexity to indicate generalization ability. For out-of-domain, we use validation data from The Pile (Gao et al., 2020), a public dataset that combines data from 22 diverse sources.

## 5 Analysis Results

In this section, we use the proposed unified view to systematically study the design choice of S-FFN. Specifically, **(1)**. we study a wide range of block sizes other than the incidental choice used in existing work and investigate its impact on language modeling perplexity (§5.1). **(2)**. Both direct and indirect block selection methods lead to lower perplexity than a standard FFN, but which type of method has better FLOPs-Perplexity trade-off and what are the relative efficacy and efficiency of different methods require further study (§5.2).

## 5.1 Memory block size

Since block size is a natural number, we aim to answer a straightforward question — given a fixed number of active memory cells $k$, **does smaller memory block size lead to lower perplexity?** We use simple and robust selection methods to disentangle the impact of hyperparameter choices. Specifically, we use random hash as recommended in HashLayer (Roller et al., 2021) (denoted **RandHash**) for indirect block selection and exact top-$k$ memory block (denoted **VanillaM**) for direct block selection. For all experiments, we use $E = 16$.

**RandHash** randomly selects $b = k/g$ unique memory blocks among all $B = d_m/g$ blocks — essentially sampling $b$ unique values from Uniform$([0, \cdots, B - 1])$. Originally, with block size $g = 4096$, a RandHash assigns a token to $4096/4096 = 1$ block; with block size $g = 2048$, $4096/2048 = 2$ blocks.

**VanillaM** originally has a block size $g = 1$ and selects top-$k$ scalars in memory coefficients $\mathbf{m} = \text{GeLU}(\mathbf{x} \cdot \mathbf{K}^\top)$. We made a minimal change to extend it to larger block size $g$: given $\mathbf{m}$, we chunk it into $B$ blocks — $\mathbf{m}^g = [\mathbf{m}^{(0)}; \cdots; \mathbf{m}^{(B-1)}]$; then, we select the top-$b$ blocks using the average of each block:

$$\text{Avg}(\text{GeLU}(\mathbf{x} \cdot (\mathbf{K}^{(i)})^\top), \text{ dim=0})^2$$

In Fig. 2, we observe that smaller block size leads to an improvement of $0.4(15.75 \rightarrow 15.35)$ perplexity for RandHash and an improvement of $0.87(15.56 \rightarrow 14.69)$ for VanillaM.

In Appendix C.2, we provide theoretical justifications for this observation which shows that a smaller block size improves model capacity by including more combinations of memory cells. For example, with $g/2$, half memory cells of expert-1 could be activated together with half of the expert-2; however, this combination is impossible with a larger block size.

| Selection method type | Method name | $g$ | $E$ | $k$ |
|---|---|---|---|---|
| Direct | VanillaM | $\{1, 64, 256, 1024, 2048, 4096\}$ (§5.1) | | |
| | LoRKM | $\{1\}$ | | |
| | PKM (Lample et al., 2019) | $\{1\}$ | $\{4, 16,$ | $\{4096,$ |
| Indirect | RandHash (Roller et al., 2021) | $\{1, 64, 256, 1024, 2048, 4096\}$ (§5.1) | $(32)^*\}$ | $(8192)^*\}$ |
| | Switch (Fedus et al., 2021) | $\{4096\}$ | | |
| | PKM-FFN (§4.1) | $\{1\}$ | | |

Table 2: All the S-FFN models used in experiments and analysis in §5 — $g$ is the number of memory cells grouped in a memory block, $k$ is the active memory cells, and $E$ control the sizes of a memory $d_m = E \cdot (4 \cdot d)$. Some settings(*) are only used for PKM.

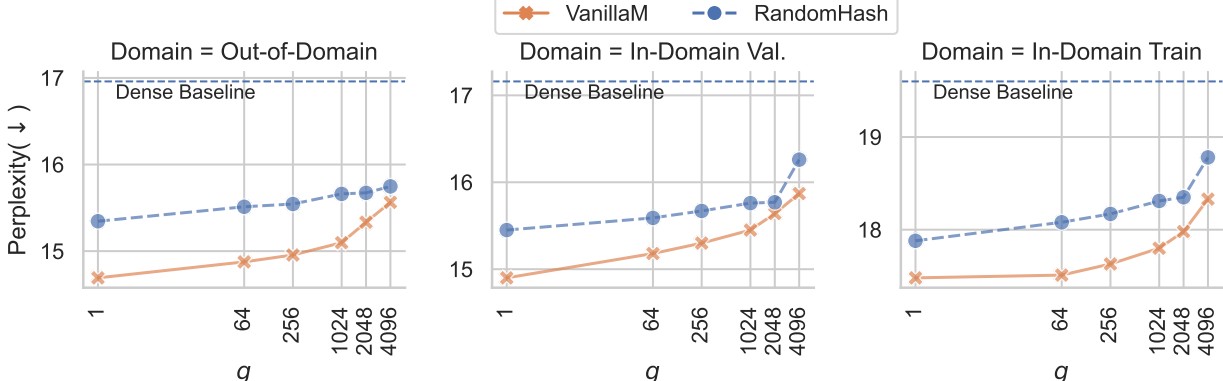

Figure 2: Perplexity (lower the better) consistently improves as memory block size $g$ decreases for both direct (VanillaM) and indirect (RandHash) selection method in S-FFN models. Ranking on individual out-of-domain test sets generally follows the ranking by average perplexity (e.g. 20 out of 22).

## 5.2 Memory block selection method

Next, we investigate the impact of the selection method, specifically, the FLOPs-perplexity trade-off for direct and indirect methods to determine the overall usefulness of each S-FFN method.

**FLOPs-perplexity trade-off** We study the efficiency of direct and indirect selection methods in S-FFN models characterized by FLOPS-perplexity trade-off. We conduct experiments across different scales of the memory by varying $E \in \{4, 16\}$; additionally, we run $E = 32$ for PKM.

In Fig. 3, we marginalize different factors used in the two selection methods — i.e. types of gates, factorization techniques on key table, etc. — and consider each type of selection method as a whole. When we change different marginalized factors, we observe that indirect methods tend to improve more as we use more FLOPs (with larger memory sizes controlled by $E$). Thus, the indirect method has a better FLOPs-perplexity trade-off.

²Avg(·) performs better than other simple aggregators — Min(·), Max(·), and Avg(| · |); see ablations in Table 7.

**Effect of gating function** We start with contrastive comparisons among PKM-FFN $^{E=16}$, PKM$^{E=32}$, RandHash$^{E=16}$ with memory block size $g = 1$ and 4096 active memory blocks. From the three parameter-matched models, we can learn important lessons to improve the design of the gate:

1. Comparing with PKM-FFN$^{E=16}$, PKM$^{E=32}$ essentially moves the parameters from a full-parameter key table to double the size of the value table.

2. PKM-FFN$^{E=16}$ and RandHash$^{E=16}$ have the same (in size) key and value tables. But the former uses a gate jointly learned with a key table, while the latter uses a learning-free gate.

As shown in Table 3, on out-of-domain, PKM-FFN$^{E=16}$ outperforms PKM$^{E=32}$(16.06) by 0.87 perplexity and slightly outperform RandHash$^{E=16}$ by 0.16. Therefore, it is essential to have **a full-parameter, and thus expressive enough, key table** to produce memory coefficients.

Table 3 shows the improvement of VanillaM$^{E=16}$, PKM-FFN$^{E=16}$, RandHash$^{E=16}$

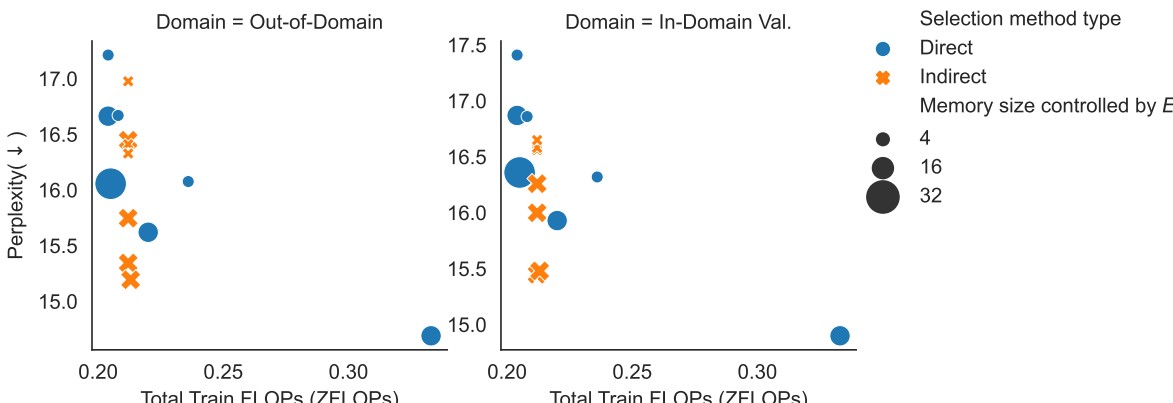

Figure 3: FLOPs-perplexity trade-off of indirect block selection is better than that of direct block selection. Indirect methods (orange cross) have more perplexity improvement relative to increases in FLOPs than direct methods (blue dots). See a more detailed legend (e.g. include methods like LoRKM) in Fig. 5.

| Selection method type | Name | # Active memory cells ($k$) | #Parameters (Entire Model) | Train ZFLOPs | Out-of-Domain Avg. ($\downarrow$) (See Table 9 for each domain) | In-Domain ($\downarrow$) | |
|---|---|---|---|---|---|---|---|
| | | | | | | Train | Val. |
| | Dense Baseline | 4096 | 354.7M | 0.212 | 16.96 | 19.60 | 17.16 |
| Direct | PKM$^{E=16}$ | 4096 | 590.2M | 0.205 | 16.66 | 19.45 | 16.87 |
| | PKM$^{E=32}$ | 4096 | 858.7M | 0.205 | 16.06 | 18.93 | 16.36 |
| | PKM$^{E=32}$ | 8192 | 858.7M | 0.213 | 16.16 | 19.05 | 16.45 |
| | VanillaM$^{E=16}$ | 4096 | 858.3M | 0.333 | **14.69** | **17.48** | **14.90** |
| Indirect | PKM-FFN$^{E=16}$ | 4096 | 858.9M | 0.213 | **15.19** | **17.82** | **15.48** |
| | RandHash$^{E=16}$ | 4096 | 858.3M | 0.212 | 15.35 | 17.88 | 15.45 |

Table 3: List of experiments for contrastively comparing designs. This table assume each memory cell is a memory block, i.e. $g = 1$. The top two best performing models (bolded) have full-parameter key table and depend more on dot product to activate parameters. Ranking on individual out-of-domain test set generally follows the ranking by average perplexity (e.g. 21 out of 22).

over Dense Baseline (16.96) are 2.27, 1.77, and 1.61 respectively on out-of-domain. They only differ by how much they depend on the key table for selection — VanillaM directly uses it, PKM-FFN indirectly gains information from it, and RandHash completely ignores it. Due to the consistent observation across out-of-domain test sets, we conclude that **the more dependent on the key table the selection method is, the better** language model it will lead to; and indirect usage (PKM-FFN) is not enough.

### 5.3 Performance of `Avg-K`

We benchmark our proposed `Avg-K` approach (§3.3) in this section.

**Language Modeling Pretraining.** Table 4 shows that the proposed S-FFN design outperforms all other indirect methods. With $< 1\%$ additional FLOPs, `Avg-K` achieves 2.16 lower perplexity than Dense Baseline (16.96), outperform Fedus et al. (2021) by 1.65 and Roller et al. (2021) by 0.5.

**Analysis** In Fig 6, although VanillaM increases its performance from block size $g = 4096$ to 256, the improvement is less significant than that of `Avg-K`. The comparison suggests that, with a larger block size, GeLU activation protects the average operation in VanillaM (after GeLU) affected by (potentially many) large negatives because $\lim_{x \to -\infty} \text{GeLU}(x) = 0$. In contrast, this "negative value" problem is mitigated by using a smaller block size, due to more blocks available for

| $E$ | #Parameters (Entire Model) | Selection method | $g$ | Train ZFLOPs | Out-of-Domain Avg. (↓) (See Table 10 for details) | In-Domain (↓) Train | Val. |
|---|---|---|---|---|---|---|---|
| 1 | 354.7M | Dense Baseline | 1 | 0.212 | 16.96 | 19.60 | 17.16 |
| 16 | ≈ 858.3M | RandHash (Roller et al., 2021) | 4096 | 0.212 | 15.75 | 18.78 | 16.26 |
| | | | 1 | 0.212 | 15.35 | 17.88 | 15.45 |
| | | Switch (Fedus et al., 2021) | 4096 | 0.212 | 16.45 | 18.20 | 16.00 |
| | | PKM-FFN | 1 | 0.213 | 15.19 | 17.82 | 15.48 |
| | | **Avg-K** | 4096 | 0.212 | 16.44 | 19.04 | 16.59 |
| | | | 256 | 0.213 | 14.91 | 17.57 | 15.19 |
| | | | 64 | 0.214 | **14.80** | **17.51** | **15.11** |

Table 4: **Avg-K** out-performs other indirect block selection methods. Switch transformer is trained with the load balancing loss to prevent model degradation (Shazeer et al., 2017; Eigen et al., 2014). Ranking on individual out-of-domain test set mostly follows the ranking by average perplexity (e.g. 21 out of 22).

selection. Since negative dot products affect **Avg-K** more, it prefers blocks with more or very positive dot products, whereas VanillaM is not shielded from extreme negatives, so it might fail to detect those blocks. Therefore, **Avg-K** could achieve an even slightly better perplexity than VanillaM for block size $g \leq 256$. See full discussions in Appendix D.2.

In Figure 7, we also include a load balancing analysis of **Avg-K**. To our surprise, the mode collapse (i.e., imbalanced usage of memory block) issue in **Avg-K** still exists as load balancing loss is not enforced. Given its superior performance, this suggests that **Avg-K** learned a good representation for each memory block despite the disadvantage of load imbalance.

## 6 Related Work

**Excluded S-FFN** Terraformer's Jaszczur et al. (2021) technique on FFN is closest to our PKM-FFN because there is a low-rank learned gate to operate on each memory cells for selection. However, we exclude this method because our framework uses all memory cells in each block, but Terraformer selects 1 cell in each memory block (see our study in Appendix E.1). In finetuning scenarios, Zhang et al. (2022b) studies the connection between FFN and SMoE by turning *trained* FFN into experts and separately learning a gate. In contrast, we focus on pretraining *from scratch*.

**Approximate Nearest Neighbour (ANN) search** One might wonder whether ANN techniques could help to search for the best key in VanillaM rather than trade the expressiveness of the key table for

efficiency. For example, one could process the un-factorized key table by ANN methods like FAISS (Johnson et al., 2021) and ScaNN (Guo et al., 2020). One successful example is applying vanilla Locality-Sensitive Hashing to Reformer (Kitaev et al., 2020). However, in our preliminary study, we found that perplexity is greatly affected by the search quality, and building a data structure after every update is expensive and hard to avoid. We leave detailed discussion to Appendix E.2.

## 7 Conclusion

We provide a unified framework for designing sparse FFN in transformers and analyze existing S-FFN methods such as MoEs in the language modeling task. Using this framework based on sparse neural memory, we found that smaller memory block (e.g. expert) size improves perplexity at the cost of slightly higher computation cost. Selection methods with gates have better FLOPs-Perplexity trade-offs than without, while the gating function in current MoEs is sub-optimal. This framework enables us to instantiate a simpler S-FFN architecture that outperforms MoEs while still being efficient in training and inference.

## Limitations

**Limitations of a smaller block size $g$** With model parallelism (Lepikhin et al., 2021), multiple GPUs contains different memory block and parallelize the calculations. If with block size $g = 4 \cdot d$, a token is only routed to 1 memory block on one device, each device doubles its chance to receive more tokens with block size $g = 2 \cdot d$. Therefore, each GPU processes more tokens and requires more

*computation time*, but we didn't measure the wall time difference in our work. Better implementations could be developed to make a smaller block size for practical usage. In addition, since each memory block has its representation stored in the gating function, the smaller block will lead to more block representation stored in the gate, e.g., more learned parameters in the learned gate and a larger table for the static gate. Although RandHash with memory block size $g = 4 \cdot d$ cost essentially the same with memory block size $g = 1$, computing $\mathbf{g}(\mathbf{x})$ for learned gates requires more cost (details in Appendix C.3.1).

As discussed in Appendix C.3.2, smaller memory block size will induce higher communication cost given the current `all_to_all`-based implementation framework (e.g. Switch Transformer). We think reducing memory block size to 1 is too extreme to be practical; and there should be a sweet spot between 1 and 4096 (or the chosen expert size) allowed by the implementation and hardware status.

**Limitations of the unified framework** Since our method `Avg-K` essentially applies an average pooling to the key table $\mathbf{K}^g$, a better alternative may exist. Our method also heavily depends on dot product information, but this might not be the best information to be used. Due to the curse of dimensionality, future work might want to focus on finding a better metric than dot product and other aggregation methods than average to measure distance between high-dimensional vectors.

Also, we didn't train `Avg-K` with load-balancing due to our current limit in budget, but we include our rationale in Appendix D.3 for why `Avg-K` should work with load-balancing.

Additionally, in large-scale SMoE training, the speed is limited by the most heavy-loaded GPU when model parallelism is used. Therefore, load balancing is essential. We also note that our scale is relatively small and does not use model parallelism, so the problem is not pronounced for us. Future follow-up should look at how to incorporate load balancing into the unified framework and inspire better actionable design choice. We think such unification requires more advanced theoretical connection with memory block and block selection method, which likely involves consideration of training procedure.

## Ethics Statements

Due to the nature of pretraining, the carbon footprint of our work is large estimated by the amount of FLOPs and GPUs reported in the paper. We did make the effort to minimize the cost at design stage of the project. In our preliminary study, we ask for recommendation from one of the authors of Artetxe et al. (2021) to choose and verify the minimal model size and amount of tokens that sufficiently differentiate different design choices.

Another ethical concern of the paper is from the pretraining data we use. As we used the same data source as Artetxe et al. (2021), we refer the reader to the ethics statements in Artetxe et al. (2021) for how much trained model absorbs bias and toxicity from training data.

## Acknowledgement

We would like to thank (in random order) helpful feedbacks from Suchin Gururangan, Xiaochuang Han, Luke Zettlemoyer, Noah Smith, pre-doctoral members of Noah's Ark, and anonymous reviewers. In developing the experimentation of this work, great help has been received from Jingfei Du, Susan Zhang, and researchers at Meta AI. We would also want to thank Zhaoheng Billy Li for his help in developing analytical formula in Appendix C.2.

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

## A Experimental details

### A.1 Hyperparameters

Table 5 specifies shared hyperparameters across all experiments, in which Table 5a contains ones for training data, optimizer, and efficient infrastructure techniques; and Table 5b for architecture. Then, Table 6a describes the hyperparameters specifically for Switch, Table 6b for LoRKM, Table 6c for PKM. In preliminary study, we train baselines with different random seeds and found the results are almost identical. Therefore, in the interest of time and carbon cost, we didn't run with different random seeds for the experiments in this paper.

### A.2 Infrastructure

We used `fairseq` as our codebase to conduct experiments. For training, all the training are done on V100 GPUs. Each training takes 32 GPUs; in rare circumstances, the number of GPU is adjusted for training speed.

### A.3 Data

Here is a detailed description of our pretraining corpus.

- **BookCorpus** (Zhu et al., 2015) consists of more than 10K unpublished books (4GB);

- **English Wikipedia**, excluding lists, tables and headers (12GB);

- **CC-News** (Nagel, 2016) contains 63 millions English news articles crawled between September 2016 and February 2019 (76GB);

- **OpenWebText** (Gokaslan and Cohen, 2019), an open source recreation of the WebText dataset used to train GPT-2 (38GB);

- **CC-Stories** (Trinh and Le, 2018) contains a subset of CommonCrawl data filtered to match the story-like style of Winograd schemas (31GB);

- **English CC100** (Wenzek et al., 2020), a dataset extracted from CommonCrawl snapshots between January 2018 and December 2018, filtered to match the style of Wikipedia (292GB).

## B Product Key Memory

As mentioned in §2.2.2, LoRKM assumes that the full key table $\mathbf{K}^\top \in \mathbb{R}^{d \times d_m}$ is composed of and approximated by a downward projection $\mathbf{D} \in \mathbb{R}^{d \times d_\ell}$ and a low-rank key table $\tilde{\mathbf{K}} \in \mathbb{R}^{d_m \times d_\ell}$,

$$\mathbf{K}^\top = \mathbf{D} \cdot \tilde{\mathbf{K}}^\top$$

Lample et al. (2019) further approximated the low-rank key table $\tilde{\mathbf{K}}$. It assumes that each low-rank key vector $\tilde{\mathbf{k}}_i \in \mathbb{R}^{d_\ell}$ is created from the concatenation from two sub-key vectors $\mathbf{c}, \mathbf{c}' \in \mathbb{R}^{\frac{d_\ell}{2}}$;

(a) Shared configuration for data, optimizer, and efficient infrastructure

| Name | Values |
| --- | --- |
| #Tokens for training | 60e9 |
| #Tokens for warm-up | $375 \cdot 1024^2$ |
| #Tokens per batch | $0.5 \cdot 1024^2$ |
| #Tokens per sample | 2048 |
| #GPU | 32 |
| GPU | NVIDIA Tesla V100 32GB |
| Optimizer | Adam($\beta$s $= (0.9, 0.98), \epsilon = 1e - 8$) |
| Weight Decay | 0.01 |
| Peak Learning Rate | 3e-4 |
| Learning Rate Scheduler | polynomial_decay |
| clip_norm | 0.0 |
| DistributedDataParallel backend | FullyShardedDataParallel |
| memory-efficient-fp16 | True |
| fp16-init-scale | 4 |
| checkpoint-activations | True |

(b) Shared configuration for architecture.

| Name | Values |
| --- | --- |
| Objective | Causal Language Model (CLM) |
| Activation function($f$) | GeLU |
| Model dimension ($d$) | 1024 |
| $d_m$ of non-S-FFN | $4 \cdot 1024$ |
| #Attention Head | 16 |
| #Layer | 24 |
| Dropout Rate | 0.0 |
| Attention Dropout Rate | 0.0 |
| share-decoder-input-output-embed | True |

Table 5: Shared configuration. This is also used for training base model.

(a) Switch

| Name | Values |
| --- | --- |
| moe-gating-use-fp32 | True |
| moe-gate-loss-wt | 0.01 i.e. CLM loss + 0.01· auxiliary loss (Fedus et al., 2021) |
| Divide expert gradients by | $\sqrt{\# \text{ Expert}} = \sqrt{B}$ |

(b) LoRKM

| Name | Values |
| --- | --- |
| $d_\ell$ | 128 |
| BatchNorm after $\mathbf{x} \cdot \mathbf{D}$ | False |

(c) PKM

| Name | Values |
| --- | --- |
| $d_\ell$ | 128 |
| # key table (§2.2.2) | 1 |
| BatchNorm after $\mathbf{x} \cdot \mathbf{D}$ | True |

Table 6: Specific architecture configuration

and the two sub-key vectors is from two smaller and non-overlapped *sub-key* tables $\mathbf{C}, \mathbf{C}' \in \mathbb{R}^{\sqrt{d_m} \times \frac{d_\ell}{2}}$.

Unlike LoRKM where key vectors are independent of each other, key vectors in PKM have some overlaps with each other and have a *structured sharing*,

$$\tilde{\mathbf{k}}_i = \left[ \mathbf{c}_{\lfloor i/\sqrt{d_m} \rfloor}, \ \mathbf{c}'_{i \ (\mathrm{mod} \ \sqrt{d_m})} \right] \in \mathbb{R}^{d_\ell}$$

One could exploit this structure to efficiently compute the memory coefficient $\mathbf{m}$. One first calculates the dot product between sub-key tables and down-projected input $\mathbf{t}$ individually and combines them with a negligible cost to form the full dot product $\mathbf{m} \in \mathbb{R}^{d_m}$:

$$m_i = f\left(\mathbf{s}_{\lfloor i/\sqrt{d_m} \rfloor} + \mathbf{s}'_{i \ (\mathrm{mod} \ \sqrt{d_m})}\right)$$

$$\text{where } \mathbf{s} = \mathbf{t}\left[: \frac{d_\ell}{2}\right] \cdot \mathbf{C}^\top \in \mathbb{R}^{\sqrt{d_m}},$$

$$\mathbf{s}' = \mathbf{t}\left[\frac{d_\ell}{2} :\right] \cdot (\mathbf{C}')^\top \in \mathbb{R}^{\sqrt{d_m}}$$

$$\mathbf{t} = \mathbf{x} \cdot \mathbf{D} \in \mathbb{R}^{\sqrt{d_\ell}}$$

## C Block size

### C.1 VanillaM with block size $g > 1$

For VanillaM with block size $g > 1$, we also tried three other simple aggregation function, but they all under-perform Average. We show their results in Table 7.

### C.2 Analysis of smaller block sizes

We first quantify the intuition —"usage of model memory is more spread out" by number of activated memory cells shared between two random tokens — $\mathbb{E}[r]$. We define this quantity to be average of every S-FFN layer, to reflect the overall behavior

$$\mathbb{E}[r] = \frac{1}{L_{\text{S-FFN}}} \sum_\ell \mathbb{E}[r_\ell]$$

where $L_{\text{S-FFN}}$ is the number of S-FFN. Because block selection usually depends on a contextualized token embedding, it's hard to draw tokens in an i.i.d. fashion. Therefore, we estimate the the quantity by evaluating the model on a validation set. We sample $N$ token pairs from each sequence for estimation:

$$\mathbb{E}[r_\ell] = \frac{1}{|\text{val}| \cdot N} \sum_{s \in \text{val}} \sum_{\substack{i=0:(x,y)_i \sim \text{Uniform}(s \times s)}}^{N-1} |\mathcal{I}_x \cap \mathcal{I}_y| \cdot g$$

where $\mathcal{I}_x$ is the indices of selected memory block for token at position $x$, and similarly for $\mathcal{I}_y$.

RandHash, though, is an exception where uniform sampling is used. Therefore, $\mathbb{E}[r]$ could also be analytically calculated for various $g$, *when assuming tokens are also uniformly distributed*.[3]

$$\mathbb{E}[r] = \frac{1}{L_{\text{S-FFN}}} \cdot L_{\text{S-FFN}} \cdot \mathbb{E}[r_\ell]$$

$$= \sum_{i=1}^b \underbrace{\binom{b}{i}}_{\substack{\text{No. of such} \\ \text{block assignments}}} \times$$

$$\underbrace{\prod_{j=0}^{i-1} \frac{b-j}{B-j}}_{\substack{\text{Probability of} \\ i \text{ overlaps}}} \cdot \underbrace{\prod_{k=0}^{b-i-1} \frac{B-b-k}{B-k}}_{\substack{\text{Probability of} \\ j \text{ non-overlaps}}} \times$$

$$\underbrace{i \cdot g}_{\substack{r \text{ cells} \\ \text{in an overlap}}}$$

In Fig. 4a, 4b, we evaluate our model with $E = 16$ on our validation subset and calculate the estimations across various $g$. **It is observed that less sharing happens as block size decreases.** However, the empirical estimation for RandHash are relatively constant across granularity. We suspect this is due to the Zipf's law of tokens. Also, we note that the magnitude of $\mathbb{E}[r]$ are different for different methods. We defer the reason of this phenomena to future work.

### C.3 Cost of smaller block sizes

### C.3.1 Cost of gate

RandHash is efficient for computation because a hash table theoretically has time complexity $O(1)$. In contrast, a conventional learned gate 2.2.1 has an $d$-dimensional embedding for each memory block. Therefore, with total of $B$ memory blocks, it has the time complexity of $O(d \cdot B)$. In Table 8 we show how the FLOPs percentage of learned gate in a single forward-backward computation changes with respect to the change in memory block size, where we assume setup in §4 is adopted.

---

[3]In the formulae, we use $\cdot$ and $\times$ interchangeably for better presentation

| Selection method | $g$ | #Parameters (Entire Model) | Train ZFLOPs | Aggregator | Out-of-Domain (22 domains; Avg. ± Std.) | In-Domain Train | Val. |
|---|---|---|---|---|---|---|---|
| Dense Baseline | 1 | 354.7M | 0.212 | N/A | 16.96 ± 5.20 | 19.60 | 17.16 |
| VanillaM | 4096 | 858.3M | 0.333 | Avg(·) | **15.56 ± 4.62** | 18.33 | **15.87** |
| | | | | Avg(\| · \|) | 15.67 ± 4.66 | — | 15.94 |
| | | | | Max(·) | 16.11 ± 4.86 | — | 16.33 |
| | | | | Min(·) | 94.86 ± 57.63 | — | 17.08 |

Table 7: VanillaM with different simple aggregators

| TFLOPs of | Memory block size ($g$) | | | | | | | | |
|---|---|---|---|---|---|---|---|---|---|
| | 4096 | 2048 | 1024 | 512 | 256 | 128 | 64 | 32 | 1 |
| 4 learned gates (across 24 layers) | 0.275 | 0.552 | 1.10 | 2.20 | 4.40 | 8.80 | 17.6 | 35.2 | 1124 |
| Entire model | 1850 | 1850 | 1860 | 1860 | 1860 | 1860 | 1870 | 1890 | 2980 |
| % | 0.0149 | 0.0298 | 0.0595 | 0.118 | 0.237 | 0.473 | 0.941 | 1.86 | 37.718 |

Table 8: FLOPs percentage of learned gate increases when memory block size $g$ decreases

### C.3.2 Cost of communication

The conventional training framework of MoE (Fedus et al., 2021) depends on `all_to_all` operations (Paszke et al., 2019) to route tokens to different devices. One might expect the communication cost remains the same if the number of device doesn't change. However, this assumes the tokens are identified by their type. In fact, the training framework further identify the routed tokens type by the experts it routed to. Therefore, the communication cost scales linearly with respect to the change in the number of memory block.

## D  Avg-K

### D.1  Rationale to use Avg in Avg-K

We heavily base our choice on experiments with aggregators in VanillaM (in Table 7). From the experiments with average absolute value (after GeLU), we hypothesized that a positive feature is good at predicting the value of a label/token against all others. In contrast, a negative value is good at negating the prediction of a single token. As such, positive features are more predictive than negative ones. Although the situation might be different for **Avg-K** (before GeLU), we expect the selection will only be affected more because of the larger impact of

negative value.

Also, we consider the experiment with max-pooled hidden states (i.e., Max(·)). This experiment shows that a memory block hardly has a single key-value cell that dominates over others since Max(·) underperforms Avg(·) and Avg(\| · \|). What makes it worse, the max operation will overlook lots of hidden states at selection, but the overlooked hidden states still contribute to the computation. In contrast, the performance increases when we consider the average (or average of the absolute values) where every hidden state contributes to the decision. Although the situation is slightly different in **Avg-K**, the "max-pooled" version of **Avg-K** will only overestimate the hidden states information even more, and the aggregated value won't be indicative of the hidden states used for computation.

The last consideration we have is that the average function is linear. When we select experts, we use the dot product between input and averaged keys. Due to the linearity, this value is equivalent to taking the dot product between the input and every key and taking the average (See Appendix D.2). Thus, using this design choice saves a great amount of computation compared with VanillaM, while keeping the neural memory analogy.

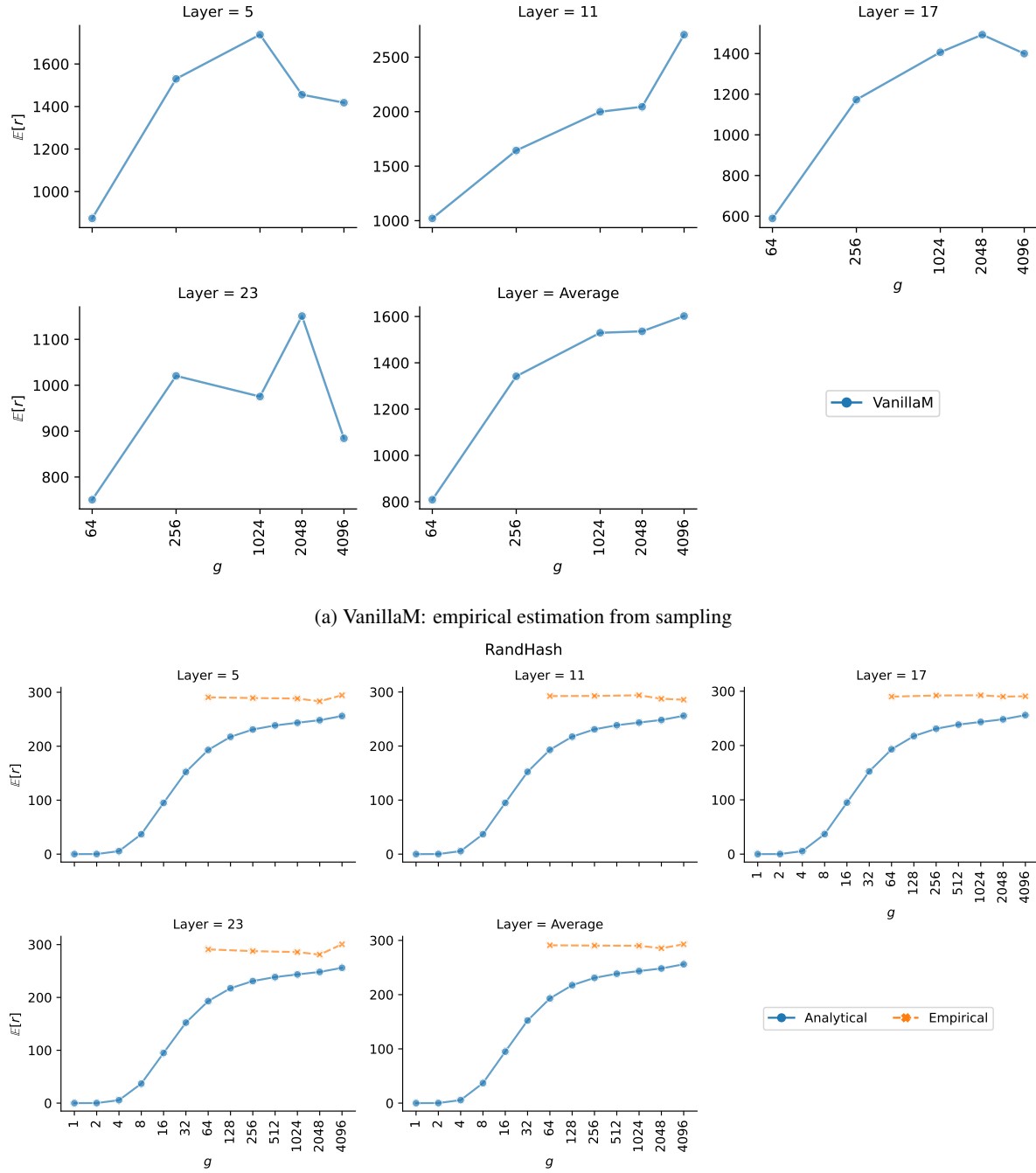

(a) VanillaM: empirical estimation from sampling

(b) RandHash: both analytical value from close form calculation and empirical value from sampling

Figure 4: Expected of shared memory cells across various block size $g$

## D.2 Avg-K analysis through comparison with VanillaM

**Avg-K** essentially applies an average pooling to the unfactorized $\mathbf{K}^g$ to create representation of each block. Due to the linearity of averaging, the operation $\mathbf{e}_i \cdot \mathbf{x}$ is equivalent to calculate the average of dot products within a block before GeLU and select blocks with the average of dot products:

$$
\begin{aligned}
\mathbf{e}_i \cdot \mathbf{x} &= \left( \frac{1}{g} \cdot \sum_{j=0}^{g-1} \mathbf{k}_j^{(i)} \right) \cdot \mathbf{x} \\
&= \frac{1}{g} \cdot \sum_{j=0}^{g-1} \left( \mathbf{k}_j^{(i)} \cdot \mathbf{x} \right) \\
&= \texttt{Avg} \left( \mathbf{x} \cdot \left( \mathbf{K}^{(i)} \right)^\top, \texttt{dim=0} \right) \quad \textbf{(Avg-K)}
\end{aligned}
$$

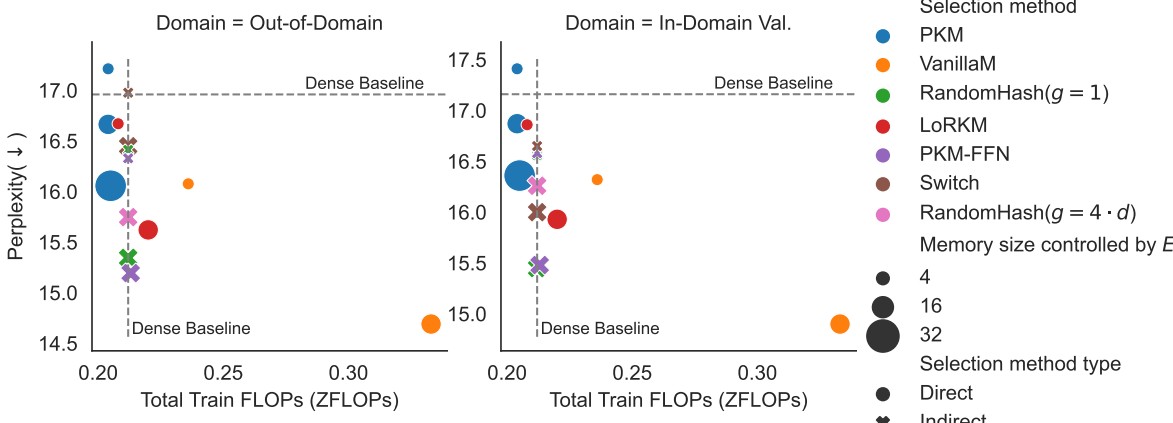

Figure 5: FLOPs-Perplexity trade-off different models where direct/indirect methods are further distinguished by model name.

In contrast, VanillaM uses average *after* GeLU(§5.1):

$$\frac{1}{g}\sum_{j=0}^{g-1} \text{GeLU}\left(\mathbf{k}_j^{(i)}\cdot\mathbf{x}\right) \quad \text{(VanillaM)}$$

Because GeLU is a non-linear function, average from `Avg-K` could be shared across tokens. In contrast, VanillaM can't, and thus making `Avg-K` efficient.

In Fig 6, we experiment both methods with various $g$. We observe when $g$ decreases from 4096, the perplexity of `Avg-K` drops more drastically than VanillaM. We believe this observation highlights the impact of GeLU. Because $\lim_{x\to-\infty} \text{GeLU}(x) = 0$, it protects the average in VanillaM from some very negative values. Thus, `Avg-K` with larger $g$ included more and potentially very negative values to average over, and thus leads to worse choices than ones made by VanillaM. On the other hand, when $g$ decreases, this "negative value" problem is mitigated. When there are more blocks available for selection (smaller $g$), because negative dot products affects `Avg-K` more, it prefers blocks with more or very positive dot products; whereas, VanillaM is protected from negative value so it fails to detect those blocks. Therefore, `Avg-K` with $g \leq 256$ could achieve an even better perplexity.

### D.3 Why `Avg-K` with load balancing should work?

Comparing VanillaM and `Avg-K`, one would expect `Avg-K` to be greatly affected by extremely negative hidden states (before GeLU). Yet, the final model with `Avg-K` could even outperform VanillaM with

the same block size (Fig. 6). This means the model will accommodate small design changes.

Additionally, the requirement of a load balancing loss is determined by the sparsity of gradients. If one "expert" gets updated with gradients while other "experts" are starved then the one expert will be selected all of the time leading to a form of mode collapse. In fact, with the same memory block size ($g = 4096$), we are surprised to observe `Avg-K` (w/o load balancing loss; row 6 in Table 4) could still perform on par with Switch (w/ load balancing; row 4 in Table 4). As our load balancing analysis suggests in Appendix D.4, when the number of experts is small, the mode collapse issue in `Avg-K` is severe. This makes us more confident that `Avg-K` will perform better with standard load balancing loss added.

Therefore, we believe that with the loss added, the model will accommodate and could still perform competitively.

### D.4 `Avg-K` load balancing analysis

On the same validation set as used in §C, we also conduct a load balancing analysis of memory blocks. Fig. 7 shows that `Avg-K` and VanillaM disproportionally used some memory blocks.

## E Preliminary study for related work

### E.1 Terraformer analysis

**Controller in Terraformer** Jaszczur et al. (2021) uses a controller to score all memory cells and pre-select a subsets — Controller($\mathbf{x}$) — for computa-

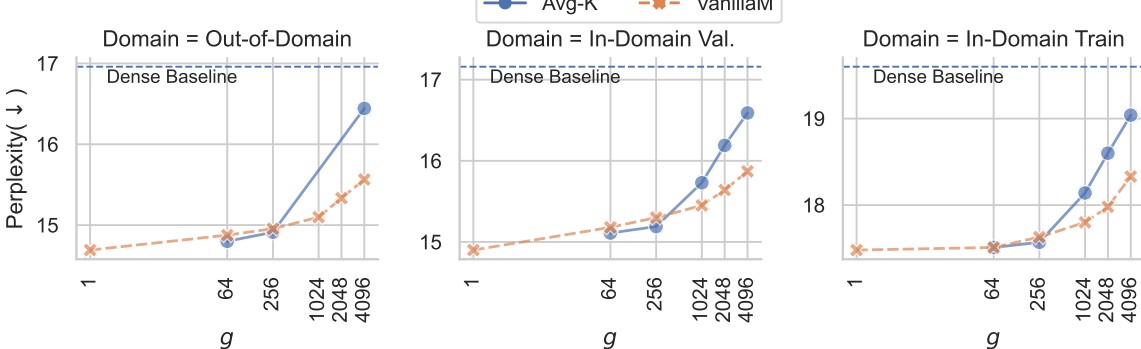

Figure 6: Perplexity performance (lower the better) of `Avg-K` and VanillaM across various $g$. We observe large drop in perplexity when $g$ decreases in `Avg-K` and less so in VanillaM; and `Avg-K` slightly outperform VanillaM with $g \leq 256$.

tion.

$$\mathbf{y} = \sum_{i \in \text{Controller}(\mathbf{x})} f(\mathbf{x} \cdot \mathbf{k}_i) \cdot \mathbf{v}_i$$

This is closest to our PKM-FFN, since their controller is essentially a gate with low-rank key table in LoRKM— $\mathbf{g}(\mathbf{x}) = (\mathbf{x} \cdot \mathbf{D}) \cdot (\mathbf{K}')^\top$, where $\mathbf{D} \in \mathbb{R}^{d \times d_\ell}$, $\mathbf{K} \in \mathbb{R}^{d_m \times d_\ell}$, and $d_\ell \ll d$. The difference is that they additionally assume the estimation from gate (and memory) could be seen as chunked into blocks and only select top-1 memory cell scored by the controller from each blocks:

$$\mathbf{y} = \sum_{i=0}^{B-1} \mathbf{g}(\mathbf{x})_{j*}^{(i)} \cdot f(\mathbf{x} \cdot \mathbf{k}_{j*}^{(i)}) \cdot \mathbf{v}_{j*}^{(i)}$$

where $j* = \arg\max_j \mathbf{g}(\mathbf{x})_j^{(i)}$. Therefore, their number of active memory cells $k$ is equal to $d_m/g$.

Similar to our contrastive pair of PKM-FFN and VanillaM, we hypothesize a "vanilla" version of their methods. Memory is chunked into blocks of size $g$ — $\mathbf{K}^g = [\mathbf{K}^{(0)}; \cdots ; \mathbf{K}^{(B-1)}]$ and similarly for $\mathbf{V}^g$. Then, one chooses the top-1 with $\mathbf{x} \cdot (\mathbf{K}^{(i)})^\top$. We call it **VanillaController**.

$$\mathbf{y} = \sum_{i=0}^{B-1} f(\mathbf{x} \cdot \mathbf{k}_{j*}^{(i)}) \cdot \mathbf{v}_{j*}^{(i)}$$

where $j* = \arg\max_j \mathbf{x} \cdot (\mathbf{K}^{(i)})^\top$. In Fig. 8, we compare VanillaController to VanillaM with $g = 1$, because the *actual* section is at the level of $g = 1$. We set $k$ in VanillaM to the one determined by equation above. We observe VanillaM outperforms VanillaController. Although the controller design

as a gating function is justified (§5.2), the decision choice of "chunking memory but only select the best memory cells" seems unmotivated. Thus, we exclude this design setup from our analysis.

### E.2 ANN

Since ANN is an approximation to exact search, we propose to randomly sabotage VanillaM, which uses the exact search. Given a $k$, we randomly swap $n\%$ of the top-$k$ of memory coefficient $\mathbf{m}$ (exact search results) with non-top-$k$ values (during training and validation), and has accuracy $(100 - n)\%$ We call it **Naive-ANN**. This is meant to set up a random baseline for ANN, because different ANN techniques might make systematic mistakes, rather than a random one. However, we believe this could still serve as a proxy and shed light on how it affects performance. As we see in Fig. 9, the model quality is sensitive to the quality of ANN.

In our preliminary study, we found building data structure after every update is expensive. This leads to some critical drawback when we apply the techniques to model parameter. Although one could amortize the cost by periodically building, the outdated data structure will lead to lower accuracy. If one chooses a hyperparameter that leads to higher quality, the cost of preprocessing and the corresponding search will be even higher. What makes it worse, the current ANN methods' search either don't support speedup by using GPU, or is not very well-integrated with GPUs — slower than calculating the exact dot product with CUDA kernel.

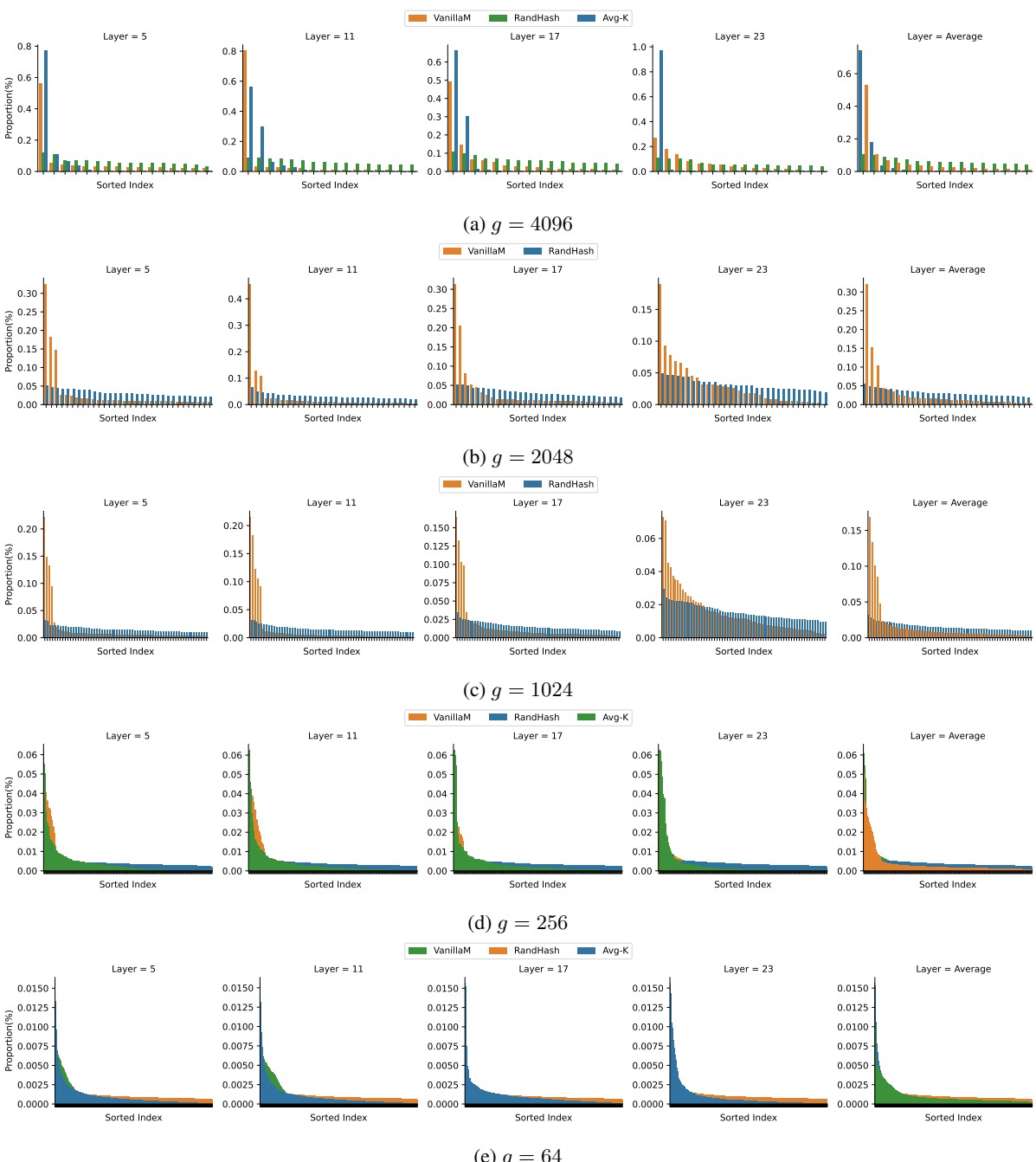

Figure 7: Load balancing of VanillaM, RandHash, **Avg-K**. The height of bar represents the proportion of memory block usage with which the memory block are sorted (in descending order).

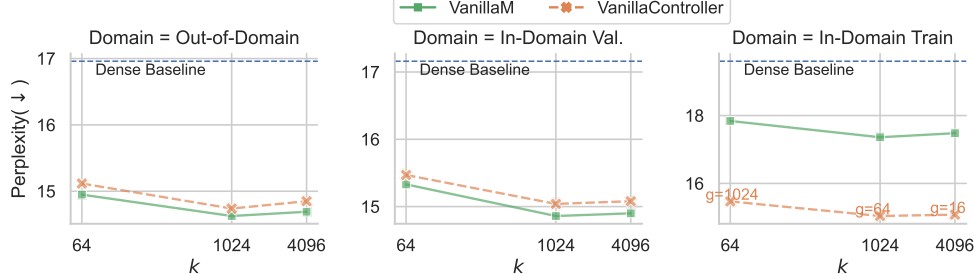

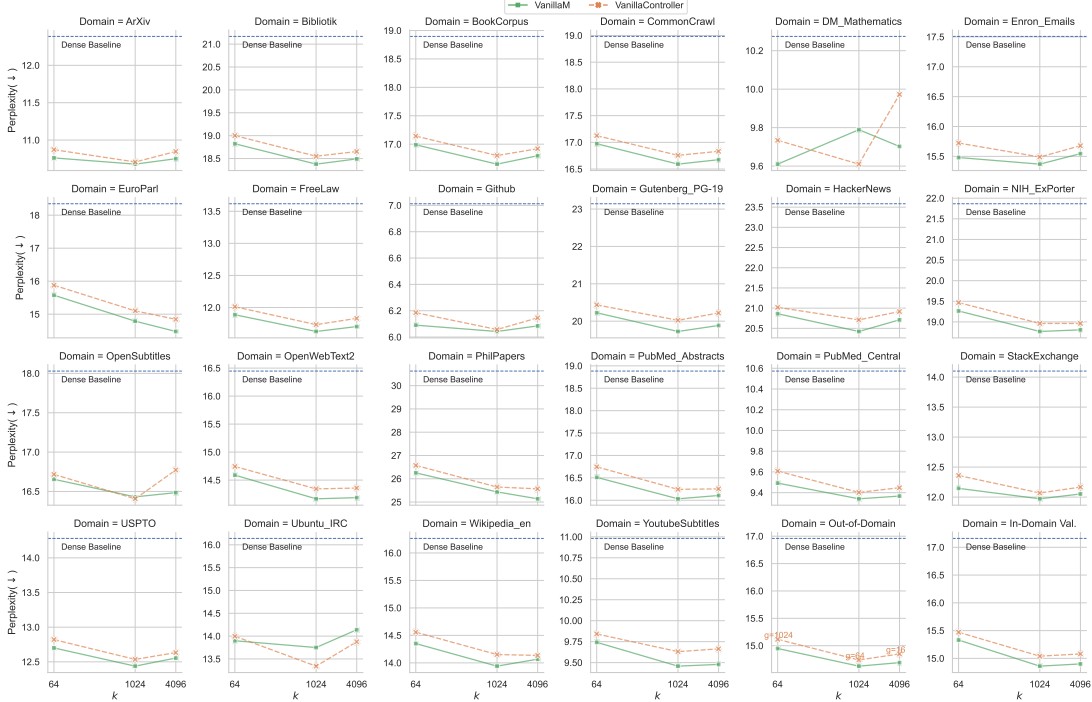

(a) Average perplexity performance (lower the better)

(b) Performance on individual domain in PILE perplexity (lower the better)

Figure 8: Perplexity performance (lower the better) of VanillaM (g=1) and VanillaController with $E = 16$.

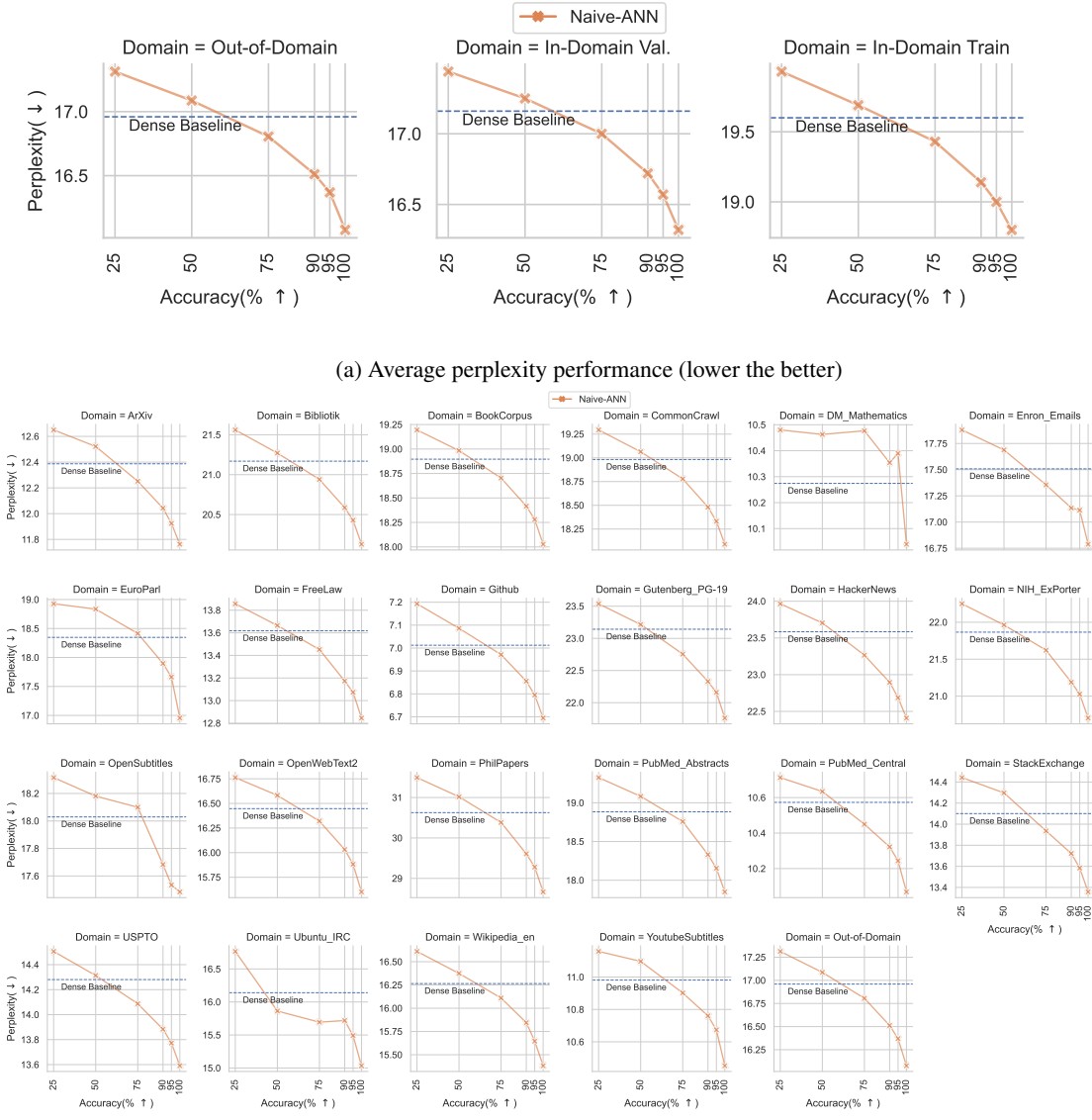

(a) Average perplexity performance (lower the better)

(b) Performance on individual domain perplexity in PILE (lower the better)

Figure 9: Perplexity performance (lower the better) of Naive-ANN accuracy with $E = 4$.

| Selection method Type | | Direct | | | | Indirect | |
|---|---|---|---|---|---|---|---|
| | | PKM | | | VanillaM | PKM-FFN | RandHash |
| Selection method | Dense Baseline | | | | | | |
| $E$ | 1 | 16 | 32 | 32 | 16 | 16 | 16 |
| $k$ | 4096 | 4096 | 4096 | 8192 | 4096 | 4096 | 4096 |
| ArXiv | 12.39 | 12.20 | 11.82 | 11.89 | **10.75** | **11.05** | 11.22 |
| Bibliotik | 21.17 | 20.82 | 20.15 | 20.25 | **18.49** | **19.11** | 19.33 |
| BookCorpus | 18.90 | 18.61 | 18.09 | 18.19 | **16.80** | **17.26** | 17.49 |
| CommonCrawl | 18.98 | 18.68 | 18.09 | 18.20 | **16.68** | **17.23** | 17.40 |
| DM_Mathematics | 10.27 | 10.34 | 10.05 | 10.28 | **9.70** | **9.72** | 9.91 |
| Enron_Emails | 17.51 | 17.23 | 16.67 | 16.68 | **15.55** | **15.90** | 16.18 |
| EuroParl | 18.35 | 17.79 | 17.01 | 17.05 | **14.48** | 15.50 | **15.03** |
| FreeLaw | 13.62 | 13.34 | 12.84 | 12.93 | **11.70** | **12.11** | 12.29 |
| Github | 7.01 | 6.91 | 6.67 | 6.68 | **6.08** | **6.25** | 6.37 |
| Gutenberg_PG-19 | 23.14 | 22.61 | 21.83 | 22.03 | **19.88** | **20.74** | 21.07 |
| HackerNews | 23.58 | 23.35 | 22.44 | 22.62 | **20.71** | **21.36** | 21.76 |
| NIH_ExPorter | 21.87 | 21.45 | 20.59 | 20.77 | **18.81** | **19.48** | 19.69 |
| OpenSubtitles | 18.03 | 17.99 | 17.46 | 17.44 | **16.48** | **16.84** | 17.10 |
| OpenWebText2 | 16.45 | 16.15 | 15.60 | 15.68 | **14.19** | **14.73** | 14.74 |
| PhilPapers | 30.63 | 30.02 | 28.50 | 28.74 | **25.14** | **26.44** | 26.60 |
| PubMed_Abstracts | 18.88 | 18.50 | 17.71 | 17.92 | **16.11** | **16.75** | 16.90 |
| PubMed_Central | 10.57 | 10.40 | 10.08 | 10.14 | **9.37** | **9.66** | 9.71 |
| StackExchange | 14.10 | 13.85 | 13.37 | 13.45 | **12.05** | **12.46** | 12.72 |
| USPTO | 14.28 | 14.07 | 13.58 | 13.68 | **12.55** | **12.96** | 13.09 |
| Ubuntu_IRC | 16.14 | 15.40 | 14.95 | 15.08 | **14.14** | **14.35** | 14.62 |
| Wikipedia_en | 16.26 | 16.03 | 15.33 | 15.48 | **14.07** | **14.51** | 14.59 |
| YoutubeSubtitles | 10.98 | 10.86 | 10.41 | 10.41 | **9.48** | 9.87 | **9.77** |
| Average | 16.96 | 16.66 | 16.06 | 16.16 | **14.69** | **15.19** | 15.35 |

Table 9: Detailed out-of-domain perplexity for Table 3. Best two performance on each domain is in bold. Relative ranking on each domain generally follows the relative ranking by averaged performance (i.e. last row).

| Selection method | Dense Baseline | RandHash | | Switch | PKM-FFN | Avg-K | | |
|---|---|---|---|---|---|---|---|---|
| $g$ | 1 | 4096 | 1 | 4096 | 1 | 4096 | 256 | 64 |
| ArXiv | 12.39 | 11.42 | 11.22 | 12.32 | 11.05 | 12.09 | 10.99 | **10.85** |
| Bibliotik | 21.17 | 19.84 | 19.33 | 19.87 | 19.11 | 20.49 | 18.70 | **18.61** |
| BookCorpus | 18.90 | 17.94 | 17.49 | 17.85 | 17.26 | 18.33 | 16.86 | **16.82** |
| CommonCrawl | 18.98 | 17.84 | 17.40 | 17.70 | 17.23 | 18.42 | 16.89 | **16.81** |
| DM_Mathematics | 10.27 | 10.22 | 9.91 | 10.63 | 9.72 | 10.25 | 9.62 | **9.61** |
| Enron_Emails | 17.51 | 16.70 | 16.18 | 17.36 | 15.90 | 17.20 | 15.65 | **15.55** |
| EuroParl | 18.35 | 15.55 | 15.03 | 19.63 | 15.50 | 17.38 | 15.32 | **15.13** |
| FreeLaw | 13.62 | 12.56 | 12.29 | 12.80 | 12.11 | 13.20 | 11.84 | **11.77** |
| Github | 7.01 | 6.51 | 6.37 | 6.96 | 6.25 | 6.89 | 6.18 | **6.12** |
| Gutenberg_PG-19 | 23.14 | 21.55 | 21.07 | 21.81 | 20.74 | 22.36 | 20.07 | **19.98** |
| HackerNews | 23.58 | 22.30 | 21.76 | 22.59 | 21.36 | 22.95 | 20.82 | **20.65** |
| NIH_ExPorter | 21.87 | 20.22 | 19.69 | 20.99 | 19.48 | 21.09 | 19.09 | **19.01** |
| OpenSubtitles | 18.03 | 17.33 | 17.10 | 17.31 | 16.84 | 17.74 | **16.62** | 16.66 |
| OpenWebText2 | 16.45 | 15.16 | 14.74 | 15.51 | 14.73 | 15.87 | 14.48 | **14.39** |
| PhilPapers | 30.63 | 27.53 | 26.60 | 30.84 | 26.44 | 29.51 | 25.90 | **25.72** |
| PubMed_Abstracts | 18.88 | 17.39 | 16.90 | 18.36 | 16.75 | 18.24 | 16.34 | **16.33** |
| PubMed_Central | 10.57 | 9.94 | 9.71 | 10.28 | 9.66 | 10.30 | 9.50 | **9.45** |
| StackExchange | 14.10 | 13.04 | 12.72 | 13.78 | 12.46 | 13.73 | 12.23 | **12.13** |
| USPTO | 14.28 | 13.41 | 13.09 | 13.54 | 12.96 | 13.89 | 12.73 | **12.62** |
| Ubuntu_IRC | 16.14 | 14.95 | 14.62 | 14.78 | 14.35 | 15.50 | 14.26 | **13.59** |
| Wikipedia_en | 16.26 | 14.98 | 14.59 | 15.67 | 14.51 | 15.73 | 14.23 | **14.12** |
| YoutubeSubtitles | 10.98 | 10.06 | 9.77 | 11.25 | 9.87 | 10.59 | 9.73 | **9.67** |
| Average | 16.96 | 15.75 | 15.35 | 16.45 | 15.19 | 16.44 | 14.91 | **14.80** |

Table 10: Detailed out-of-domain perplexity for Table 4. Best performance on each domain is in bold. Relative ranking on each domain generally follows the relative ranking by averaged performance (i.e. last row).