# OpenReview forum: "Towards A Unified View of Sparse Feed-Forward Network in Pretraining Large Language Model"
_EMNLP/2023/Conference — EMNLP 2023 Main_

### Official Review · Reviewer_J9vn · 2023-08-03

**Soundness:** 3

**Excitement:**

4: Strong: This paper deepens the understanding of some phenomenon or lowers the barriers to an existing research direction.

**Paper Topic And Main Contributions:**

Topic: Large and sparse feed-forward networks S-FFN for pretraining large language models.

Main contribution: This paper analyzes the design choices of S-FFN, including the memory block size and selection method, under a general conceptual framework of sparse neural memory. The authors compare several S-FFN architectures for language modeling and provide insights into their relative efficacy and efficiency. They propose a simpler selection method, Avg-K, that selects blocks through their mean aggregated hidden states, which achieves lower perplexity in language modeling pretraining compared to existing MoE architectures. This work provides a better understanding of S-FFN and its potential for scaling up Transformers model size for pretraining large language models.

**Reasons To Accept:**

This paper systematically analyzes the role of S-FFN in large-scale language models and summarizes it into a unified design framework. After thorough experimental analysis, the authors propose a superior Avg-K method that surpasses common methods such as MoE. This provides insightful considerations for the larger parameterization and training of LLMs.

**Reasons To Reject:**

This paper provides a unified S-FFN framework from a purely theoretical and model design perspective, but lacks integration with engineering implementation. Given that large-scale pretraining is a complex system architecture, it is suggested to further explore the potential of the Avg-K method to improve efficiency and effectiveness under the premise of model and data parallelism.

**Reproducibility:**

4: Could mostly reproduce the results, but there may be some variation because of sample variance or minor variations in their interpretation of the protocol or method.

**Reviewer Confidence:**

2: Willing to defend my evaluation, but it is fairly likely that I missed some details, didn't understand some central points, or can't be sure about the novelty of the work.

---

> ### Author Rebuttal · Authors · 2023-08-28
>
> **Q1:** This paper [...] from a purely theoretical and model design perspective, but **lacks integration with engineering implementation**. Given [...] a complex system architecture, it is suggested to further explore the potential of the Avg-K [...] **under the premise of model and data parallelism**.
>
> **A1:**
>
> We extend our gratitude to the reviewer for addressing concerns related to engineering implementation. Nevertheless, we wish to clarify that **the scope of our work unfortunately does not include the communication cost.**
>
> During the initial project scoping, we observed historical parallels where the start of the proof-of-concept for MoE in neural networks was established in [1], followed by the engineering implementation for practical efficiency in [2]. Drawing a parallel, our work follows a similar trajectory — we introduce and demonstrate the potential of a (theoretically) unified framework to shape S-FFN design. Consequently, future research could concentrate on bridging the gaps (practically) to escalate the impact. This would entail delving into the analysis of communication costs and proposing corresponding solutions, both of which hold the potential for dedicated papers.
>
> Here are some of our considerations:
>
> Each MoE variant may require distinct implementation considerations to obtain near-optimal runtime. For instance, we thought of an optimization strategy that becomes relevant as memory block sizes decrease, leading to increased communication costs. In such a context, where multiple small memory blocks coexist within a single GPU, an intuitively efficient "local routing" program could be employed to cleverly allocate tokens among these local blocks. In contrast, the currently standard all-to-all implementation could inadvertently lead to a large number of duplicate tokens being dispatched, thereby accounting for the elevated communication cost. Implementing such strategies could potentially involve refining the underlying data structure or even creating a specialized CUDA kernel.
>
> We leave the above to future work due to access to limited resources.
>
> **References:**
>
> [1] Noam Shazeer et. al. Outrageously Large Neural Networks: The Sparsely-Gated Mixture-of-Experts Layer https://arxiv.org/abs/1701.06538
>
> [2] Dmitry Lepikhin et. al. GShard: Scaling Giant Models with Conditional Computation and Automatic Sharding https://arxiv.org/abs/2006.16668

---

### Official Review · Reviewer_mxPW · 2023-08-05

**Soundness:** 4

**Excitement:**

4: Strong: This paper deepens the understanding of some phenomenon or lowers the barriers to an existing research direction.

**Paper Topic And Main Contributions:**

This paper provides a unified mathematical view of sparse mixture-of-experts and sparse neural memory. Based on this view and empirical observations, the author proposes a new gating method Avg-k to select memory blocks based on the averaged hidden states of each block. Avg-k achieves substantially better perplexity compared with a strong baseline (Switch Transformer).

**Reasons To Accept:**

1. Strong perplexity improvement over Switch Transformer on out-of-domain evaluation of language modeling.
2. The paper is well-written and easy to follow.
3. Comprehensive analyses for the hyper-parameters and the baselines. The analyses deepen the understanding of current MoE methods.

**Reasons To Reject:**

1. While the method provides better FLOPs-perplexity trade-off, it is unclear how Avg-K can provide better actual training speed perplexity trade-off due to increased communication cost.
2. It will be better to compare the performance of the proposed method with more recent MoE routing approach such as X-MoE.

**Reproducibility:**

4: Could mostly reproduce the results, but there may be some variation because of sample variance or minor variations in their interpretation of the protocol or method.

**Reviewer Confidence:**

3: Pretty sure, but there's a chance I missed something. Although I have a good feel for this area in general, I did not carefully check the paper's details, e.g., the math, experimental design, or novelty.

---

> ### Author Rebuttal · Authors · 2023-08-28
>
> **Q1:** While [...] better FLOPs-perplexity trade-off, it is **unclear how Avg-K can provide better actual training speed** perplexity trade-off due to increased communication cost.
>
> **A1:**
>
> Avg-K on its own does not increase communication cost by much from Switch Transformer. During training, the increased communication cost of Avg-K comes from sending the averaged key of each expert to all other GPUs. In our experiments, this means, for each batch, the additional communication cost is #num_GPU x 1024 x #experts floating points. Usually this is less than one million FLOPs. According to Table 8 in appendix, this cost is negligible. During inference, when computing token-by-token generations, our selection method is cheap because the average can be pre-computed and thus reused.
>
> So in short, Avg-K only has slightly higher communication cost than baselines and is very efficient during inference.
>
>
> **Q2:** It will be better to **compare [...] with a more recent MoE routing approach such as X-MoE**.
>
> **A2:**
>
> We thank the reviewer for raising this concern. We indeed noticed X-MoE during the project, but there are a few reasons that we are not able to compare:
> 1. X-MoE is a contemporary work towards the end of this project. Nevertheless, the even harder part for adding X-MoE to our experiment is that they didn’t release their code. Therefore, unfortunately, we are not able to compare shoulder-to-shoulder with them.
> 2. In general, there are many S-FFN variants and thus it is hard to cover them all by nature. Even for [1] where they have much more GPU than us, the analysis focuses on 4 variants. In our work, we tried our best to broaden the coverage and covered 6 variants (of various #expert, see Table 2) for analysis. In addition, we also include some side experiments when we explore related work (see discussion in Sec 6 and details in Appendix D).
>
> **References:**
>
> [1] Aidan Clark, Diego de las Casas, Aurelia Guy, Arthur Mensch, et. al. Unified Scaling Laws for Routed Language Models. https://arxiv.org/abs/2202.01169

---

### Official Review · Reviewer_mMBN · 2023-08-10

**Soundness:** 4

**Excitement:**

4: Strong: This paper deepens the understanding of some phenomenon or lowers the barriers to an existing research direction.

**Paper Topic And Main Contributions:**

In this paper, the authors propose an interesting question that lies in the essence of the S-FFN approaches. The authors establish a unified framework to characterize various S-FFN approaches, which feature two key dimensions - memory block size and memory block selection method. With the elaborate analysis, the authors demonstrate a single gate for S-FFN - Avg-K in both sparse neural memory and MoE. The extensive evaluation shows the proposed Avg-K outperforms the state-of-the-art approaches.

**Reasons To Accept:**

1. The authors propose an interesting research problem on the key design choices in different S-FFN approaches.

2. The paper presents a comprehensive analysis of different S-FFN approaches and a unified framework for the key design choices.

3. The authors propose a new selection method (Avg-K) and demonstrate its performance benefits through extensive experiments.

**Reasons To Reject:**

1. For a comprehensive analysis of MoE, it is very important to explore the communication overhead among the experts (all to all).

2. For the new selection method (Avg-K), do you have any insights that guide you the determine this approach? It will be better if there is a deeper exploration.

**Reproducibility:**

4: Could mostly reproduce the results, but there may be some variation because of sample variance or minor variations in their interpretation of the protocol or method.

**Reviewer Confidence:**

3: Pretty sure, but there's a chance I missed something. Although I have a good feel for this area in general, I did not carefully check the paper's details, e.g., the math, experimental design, or novelty.

---

> ### Author Rebuttal · Authors · 2023-08-28
>
> **Q1:** For a comprehensive analysis of MoE, it is very important to explore the **communication overhead among the experts (all to all)**.
>
> **A1:**
>
> Since we have a large variety of implementations it is difficult to make a fair comparison between communication overhead between models since the overhead is mostly dependent on how optimized the implementation is. For example, we thought about blocking that is partitioned on a single node or GPU, such that there could be a “local routing” program to send tokens to different memory blocks efficiently which is much more efficient than all-to-all communication. However, doing the full optimizations for all implementations and block sizes is out of the scope of our work.
>
> We agree this is an important limitation of our work as we discussed in the limitation section(Line 552-572). We want to additionally comment that, historically, the proof-of-concept of MoE in neural networks started with [1] and the engineering implementation that makes MoE more accessible and practical (directly) started with [2]. We think similar things happen here — analyzing the communication cost and proposing solutions to the problem might each deserve the work of a paper.
>
> **Q2:** For the new selection method (Avg-K), do you have **any insights that guide you** to determine this approach? It will be better if there is a deeper exploration.
>
> **A2:**
>
> Yes. We summarize our insights in Line 300-308. Specifically, we found that the gate design of MoE provides a better performance-trainFLOPs trade-off (see Fig 3), so we need a gate. In sec 5.2, the contrastive comparison of different gate designs controls how “directly” the input vector and key table parameter can see each other’s information — from direct dot product (most direct) to hash table (least direct). In Line 473-474, we wrote “the more dependent on the key table the selection method is, the better”. And we also found that having a full-parameter key table is important (Line 450-463).
>
> Therefore,
> 1. Avg-K has a gate design;
> 2. We use aggregated key vector to dot product with input vector to make it as direct as possible;
> 3. We didn’t apply any factorization trick to the experts’ key tables.
>
> We will include these details in the updated draft.
>
>
> **References:**
>
> [1] Noam Shazeer et. al. Outrageously Large Neural Networks: The Sparsely-Gated Mixture-of-Experts Layer https://arxiv.org/abs/1701.06538
>
> [2] Dmitry Lepikhin et. al. GShard: Scaling Giant Models with Conditional Computation and Automatic Sharding https://arxiv.org/abs/2006.16668

---

### Official Review · Reviewer_M6uy · 2023-08-12

**Soundness:** 3

**Excitement:**

3: Ambivalent: It has merits (e.g., it reports state-of-the-art results, the idea is nice), but there are key weaknesses (e.g., it describes incremental work), and it can significantly benefit from another round of revision. However, I won't object to accepting it if my co-reviewers champion it.

**Paper Topic And Main Contributions:**

The reviewed paper focuses on the effectiveness and efficiency of large and sparse feed-forward networks (S-FFN), specifically Mixture-of-Experts (MoE). The study examines the two critical aspects of S-FFN: memory block size and memory block selection method, within a comprehensive framework of sparse neural memory.
The researchers compare various S-FFN architectures for language modeling, shedding light on their relative effectiveness and efficiency. They propose a unified framework to analyze the impact of memory block size and selection method. The central idea revolves around the notion of sparse neural memory, emphasizing the activation of specific FFN parameters based on input conditions.
The paper delves into the intricate design choices of S-FFN architectures, particularly MoE, showcasing how memory block size and selection methods influence model performance and efficiency.

**Questions For The Authors:**

-  Does the proposed Analysising framework cover some new variations or combinations of architectural choices?

**Reasons To Accept:**

- Conceptual Framework: The paper introduces a unified framework of sparse neural memory, which provides an  innovative insight for analyzing the impact of memory block size and selection methods in S-FFN architectures.

- The paper is generally clearly-written.

**Reasons To Reject:**

- Scope of Analysis: The paper primarily focuses on the impact of memory block size and selection methods within the context of S-FFN architectures, particularly Mixture-of-Experts (MoE). However, it might not cover all potential variations or combinations of architectural choices. Espeecially,  the research landscape in natural language processing evolves rapidly. The paper's findings and conclusions might become dated if not updated or considered in the context of subsequent advancements in the field.

- Scalability: The paper does not extensively explore the scalability of the proposed approach to even larger models or more complex language tasks. As model sizes and task complexities continue to grow, the scalability of the \textbf{\texttt{Avg-K}} method might need to be scrutinized further.



**Reproducibility:**

3: Could reproduce the results with some difficulty. The settings of parameters are underspecified or subjectively determined; the training/evaluation data are not widely available.

**Reviewer Confidence:**

3: Pretty sure, but there's a chance I missed something. Although I have a good feel for this area in general, I did not carefully check the paper's details, e.g., the math, experimental design, or novelty.

---

> ### Author Rebuttal · Authors · 2023-08-28
>
> We appreciate M6uy recognizing our S-FFN framework as an innovative contribution and highlighting our clear writing. We provide further clarifications and context to address M6uy’s concerns about the scope and impact of our work.
>
>
> **Q1: Scope of Analysis**: The paper primarily focuses on the impact of memory block size and selection [...]. However, it **might not cover all potential variations or combinations** of architectural choices. Especially, the research landscape in natural language processing evolves rapidly. The paper's findings and conclusions **might become dated if not updated or considered in the context of subsequent advancements** in the field.
>
> **A1:**
>
> With finite resources, it is not feasible for us to **cover all potential variations or combinations** of architectural choices. Therefore, we chose to focus on two key design choices of S-FFN, namely memory block size and selection method, which are widely used in sparse Transformers but yet lack thorough understanding. We conduct comprehensive ablations covering commonly used hyperparameters specific to these two design choices as well as their combinations. The empirical results are complementary to existing work on Mixture-of-Experts (MoEs) which usually only focus on one specific selection method.
>
> Regarding your concern that this work may be dated, we want to emphasize that focusing on understanding the vanilla FFN layer instead of a special instantiation makes our work stay relevant to all Transformer architectures which incorporate sparsity in the FFN layers. Given that S-FFN has been actively studied and its potential for scaling [3], we believe this work is a timely contribution to the field that provides complementary knowledge to existing work on S-FFN.
>
> **Q2: Scalability**: The paper **does not [...] explore [...] larger models or more complex language tasks**. [...] the scalability of the \textbf{\texttt{Avg-K}} method might need to be scrutinized further.
>
> **A2:**
>
> > Larger models
>
> Our experiments cover a representative range of scales — from 355M to 859M
>  by parameter size and from 0.212 to 0.333 by ZFLOPs (Table 3). While we agree that this experiment cannot be extrapolated to models as large as Switch [2] or GShard [4], we think the scaling curve we provide gives the initial evidence that S-FFN with gate design achieves much better FLOPs-perplexity trade-off than S-FFN without gate (Line 429-443, Figure 3&5).
>
> Additionally, the reason for the chosen base model size is two-fold:
> 1. At the time of development, we followed  [1, 2] to determine our base model size, **the smallest standard size for base architecture in [1, 2] is about 222M**. When designing our experiment, we chose the base size of 355M, similar to the community-estimated size of OpenAI text-ada-001.
> 2. Also informed by Fig 2 in [3], 355M is the smallest size that separates the performances of different architecture designs.
>
> We certainly hope there will be more future work on the scaling behavior of our unified framework. Thanks for pointing out the inclarity on choice of size, we will clarify this in the camera-ready version.
>
> > More complex language tasks
>
> For the evaluation of models, we found it **conventional to primarily use perplexity (i.e. auto-regressive language modeling task)** when developing or exploring model architecture design [1, 2, 3, 4, 5, 6, 7]. To gain more insight into the generalization capability, we also included the PILE datasets [9] to demonstrate the model's performance on text from 22 diverse domains. Additionally, considering the strong capabilities of auto-regressive/decoder models like GPT-3, “complex” tasks like question answering, coreference resolution, etc. could be unified by properly prompting the language model. This approach exhibits a strong correlation with the model's perplexity performance (e.g. Fig 2&4 in [8]). Therefore, as it's impractical to run every evaluation metrics, we believe our evaluation setup is sufficient for the purpose of our work.
>
> ---
>
> > Does the proposed Analysis framework **cover some new variations or combinations** of architectural choices?
>
> Yes, we included a new MoE design called Avg-K (details in Sec 3.3 and 5.3) with practical potential. This can be seen as a variation of regular MoE models where the routing mechanism is changed. In addition, we also included a new design called PKM-FFN (see more details starting Line 345, and Line 444) as a new variation only intended for analysis purposes.
>
>
> **References:**
>
> [1] Roller, Stephen, Sainbayar Sukhbaatar, and Jason Weston. "Hash layers for large sparse models." https://arxiv.org/pdf/2106.04426.pdf
>
> [2] William Fedus, Barret Zoph, and Noam M. Shazeer. Switch transformers: Scaling to trillion parameter models with simple and efficient sparsity. https://jmlr.org/papers/volume23/21-0998/21-0998.pdf
>
> [3] Aidan Clark, Diego de las Casas, Aurelia Guy, Arthur Mensch, et. al. Unified Scaling Laws for Routed Language Models. https://arxiv.org/abs/2202.01169
>
> [4] Dmitry Lepikhin et. al. GShard: Scaling Giant Models with Conditional Computation and Automatic Sharding https://arxiv.org/abs/2006.16668
>
> [5] Mike Lewis, Shruti Bhosale, Tim Dettmers, Naman Goyal, Luke Zettlemoyer. BASE Layers: Simplifying Training of Large, Sparse Model
>
> [6] Suchin Gururangan, Mike Lewis, Ari Holtzman, Noah A. Smith, Luke Zettlemoyer. DEMix Layers: Disentangling Domains for Modular Language Modeling
>
> [7] Ofir Press, Noah A. Smith, Mike Lewis Train Short, Test Long: Attention with Linear Biases Enables Input Length Extrapolation
>
> [8] Mikel Artetxe, et al. Efficient Large Scale Language Modeling with Mixtures of Experts
>
> [9] Leo Gao et. al.The Pile: An 800GB Dataset of Diverse Text for Language Modeling

---

### Meta-Review · Area_Chair_JccQ · 2023-09-20

**Recommendation:** 4

**Metareview:**

This paper analyzes design decisions for sparse feedforward layers in large language models. It presents a unified framework that connects sparse mixtures of experts with sparse neural memory and empirically compares the two methods along with new variants that take elements of each. Inspired by their analysis, the authors present a new routing method called avg-k that outperforms previous methods.

Reviewers generally found the paper clear and appreciated the strong empirical results, comprehensive analysis, and insights from the new conceptual framework. However, a central concern with the paper brought up by all reviews is potential lack of scalability to settings where model parallelism is necessary (models in the paper are up to 850M parameters and are not trained with model parallelism):
* The finding that a small block size (essentially activating a larger number of smaller experts) works better given the same compute budget may not hold because communication costs will increase.
* The lack of load balancing in Avg-K will become more of a problem because speed will be limited by the devices receiving more tokens. While the paper does argue that load balancing should combine well with avg-k, there aren’t experiments (even at smaller scale) to justify this.
* Possibly due to the lack of load balancing, avg-k works poorly compared to baselines when using a larger block size, but that might be the more realistic setting when thinking about a model-parallel version.

In the rebuttal, the authors argue that while they believe there exist implementation details or method variants that could reduce communication overhead, the engineering required to develop an efficient model-parallel implementation would be very high. Instead, they see the value of the work in presenting their unified framework and in providing findings (e.g that Avg-K with a small block size is FLOP-efficient) that could be built upon in future engineering-focused research (e.g. adding load balancing and reducing communication overhead). While I do think the limitations may diminish the impact of the paper, I don’t think we should limit research on language model sparsity only to researchers with huge engineering budgets, and I believe the findings still could be of interest to the community.

---

### Decision · Program_Chairs · 2023-10-07

**Decision:**

Accept-Main

**Comment:**

This paper analyzes design decisions for sparse feedforward layers in large language models. It presents a unified framework that connects sparse mixtures of experts with sparse neural memory and empirically compares the two methods along with new variants that take elements of each. Inspired by their analysis, the authors present a new routing method called avg-k that outperforms previous methods.

Reviewers generally found the paper clear and appreciated the strong empirical results, comprehensive analysis, and insights from the new conceptual framework. However, a central concern with the paper brought up by all reviews is potential lack of scalability to settings where model parallelism is necessary (models in the paper are up to 850M parameters and are not trained with model parallelism):
* The finding that a small block size (essentially activating a larger number of smaller experts) works better given the same compute budget may not hold because communication costs will increase.
* The lack of load balancing in Avg-K will become more of a problem because speed will be limited by the devices receiving more tokens. While the paper does argue that load balancing should combine well with avg-k, there aren’t experiments (even at smaller scale) to justify this.
* Possibly due to the lack of load balancing, avg-k works poorly compared to baselines when using a larger block size, but that might be the more realistic setting when thinking about a model-parallel version.

In the rebuttal, the authors argue that while they believe there exist implementation details or method variants that could reduce communication overhead, the engineering required to develop an efficient model-parallel implementation would be very high. Instead, they see the value of the work in presenting their unified framework and in providing findings (e.g that Avg-K with a small block size is FLOP-efficient) that could be built upon in future engineering-focused research (e.g. adding load balancing and reducing communication overhead). While I do think the limitations may diminish the impact of the paper, I don’t think we should limit research on language model sparsity only to researchers with huge engineering budgets, and I believe the findings still could be of interest to the community.